# Methane dynamics in the Baltic Sea: investigating concentration, flux and isotopic composition patterns using the coupled physical-biogeochemical model BALTSEM-CH4 v1.0

Erik Gustafsson[1,2], Bo G. Gustafsson[1,2], Martijn Hermans[2,3], Christoph Humborg[2], Christian Stranne[2,4,5]

5   [1]Baltic Nest Institute, Stockholm University, 106 91 Stockholm, Sweden
[2]Baltic Sea Centre, Stockholm University, 106 91 Stockholm, Sweden
[3]Environmental Geochemistry Group, Faculty of Science, University of Helsinki, Helsinki 00560, Finland
[4]Department of Geological Sciences, Stockholm University, 106 91 Stockholm, Sweden
[5]Bolin Centre for Climate Research, Stockholm University, 106 91 Stockholm, Sweden

10   *Correspondence to*: Erik Gustafsson (erik.gustafsson@su.se)

**Abstract.** Methane ($CH_4$) cycling in the Baltic Sea is studied through model simulations that incorporate the stable isotopes of $CH_4$ ($^{12}C$-$CH_4$ and $^{13}C$-$CH_4$) in a physical-biogeochemical model. A major uncertainty is that spatial and temporal variations of the sediment source are not well known. Further, the coarse spatial resolution prevents the model to resolve shallow-water near-shore areas for which measurements indicate occurrences of considerably higher $CH_4$ concentrations and emissions compared to the open Baltic Sea. A preliminary $CH_4$ budget for the central Baltic Sea (the Baltic Proper) identifies benthic release as the dominant $CH_4$ source, which is largely balanced by oxidation in the water column and to a smaller degree by outgassing. The contributions from river loads and lateral exchange with adjacent areas are of marginal importance. Simulated total $CH_4$ emissions from the Baltic Proper correspond to an average ~1.5 mmol $CH_4$ $m^{-2}$ $y^{-1}$, which can be compared to a fitted sediment source of ~18 mmol $CH_4$ $m^{-2}$ $y^{-1}$. A large-scale approach is used in this study, but the parametrizations and parameters presented here could also be implemented in models of near-shore areas where $CH_4$ concentrations and fluxes are typically substantially larger and more variable. Currently, it is not known how important local shallow-water $CH_4$ hotspots are compared to the open water outgassing in the Baltic Sea.

## 1 Introduction

Methane is the second-most important greenhouse gas after carbon dioxide ($CO_2$), contributing about 20% of the total radiative forcing (Etminan et al., 2016). Using top-down approaches (atmospheric observations and inverse modeling), the present-day global $CH_4$ emissions have been estimated to be 576 Tg $CH_4$ $y^{-1}$ (range 550-594), whereas bottom-up approaches (process-based modeling of land surface emissions and data on anthropogenic emissions) yield a total of 737 Tg $CH_4$ $y^{-1}$ (range 594-881; Saunois et al., 2020). The causes of the discrepancy between the two methods are not well known, but is believed to mainly reflect uncertainties in estimates of natural emissions – in particular from wetlands, lakes, and running waters (Saunois et al., 2020). The global mean atmospheric $CH_4$ level has increased by about 1000 ppb over the last two centuries (Ferretti et al., 2005). Projections of future development ranges from a gradual decrease to a massive increase depending on the development of anthropogenic emissions (Saunois et al., 2020).

The isotopic composition of atmospheric $CH_4$ ($\delta^{13}C_{CH4a}$) varies seasonally and over longer time-scales (Ferretti et al., 2005; Lan et al., 2021). Long-term trends of $\delta^{13}C_{CH4a}$ depend on the relative contributions from three main sources: biogenic (-110‰ to -50‰; e.g., wetlands -60‰), fossil (-40‰), and pyrogenic/biomass burning (-25‰ or -12‰; depending on pathways of carbon fixation in plants). Over the 20th century, a long-term increase of $\delta^{13}C_{CH4a}$ from -49‰ to -47‰ has occurred (Ferretti et al., 2005). However, a recent increase of the atmospheric $CH_4$ level has been accompanied by a decrease in $\delta^{13}C_{CH4a}$, for reasons that are not fully understood (Lan et al., 2021). The observed $\delta^{13}C_{CH4a}$ development can help to constrain different $CH_4$ sources and thus reduce their uncertainties.

It has been estimated that approximately half of the total $CH_4$ emissions come from aquatic ecosystem sources, dominated by inland water ecosystems (Rosentreter et al., 2021). The total oceanic $CH_4$ emissions, including diffusive and bubble-driven ebullitive fluxes, constitute a relatively small fraction amounting to ~6–12 Tg $CH_4$ $y^{-1}$ (Weber et al., 2019). Methane formation in sediments can be substantial, but aerobic and anaerobic oxidation processes can efficiently remove $CH_4$ both in the pore water and water column. For that reason, near-shore areas (0–50 m water depth), shallow enough to allow $CH_4$ to escape to the atmosphere before being oxidized, dominate the oceanic emissions despite representing a comparatively minor area (Weber et al., 2019). In shallow, organic-rich sediments, seafloor ebullition will increase in response to ocean warming due to increased biogenic $CH_4$ production and decreased $CH_4$ solubility (Borges et al., 2016). This notion was qualitatively supported by acoustic observations of outgassing from the sediments during a recent field study, where exceptionally high $CH_4$ emissions were reported from the coastal Baltic Sea at the end of a summer heat wave (~250 μmol $m^{-2}$ $d^{-1}$, Humborg et al., 2019).

The coastal ocean is currently a net $CO_2$ sink, which depending on method (observations or model calculations) has been estimated to approximately 0.44-0.72 Pg C $y^{-1}$ (Resplandy et al., 2024). Emissions of the powerful greenhouse gases nitrous oxide ($N_2O$) and $CH_4$ can, however, offset the $CO_2$ uptake in the net radiative balance of the coastal ocean: while highly uncertain, preliminary estimates indicate an offset in a range 30-60% (Resplandy et al., 2024). These numbers highlight the crucial importance of more accurate estimates of both $N_2O$ and $CH_4$ fluxes from coastal areas when determining the influence of the coastal ocean on climate.

In the Baltic Sea, there are strong gradients in $CH_4$ concentrations both from near-shore areas to open Baltic Sea surface waters (e.g., Gülzow et al., 2013; Humborg et al., 2019) and from surface to deep water (e.g., Schmale et al., 2010; Jakobs et al., 2013). Substantial parts of Baltic Sea deep waters are stagnant over extended periods in time, which in combination with high loads of organic material cause episodic anoxia (e.g., Carstensen et al., 2014). During stagnant anoxic periods, $CH_4$ accumulates and reaches concentrations ranging from 1000 to 3000 nM (Jakobs et al., 2013; 2014; Ketzer et al., 2024). This $CH_4$ is, however, largely consumed by aerobic oxidation processes (MOX) when mixed into the redoxcline at intermediate depths (Jakobs et al., 2013). Peak oxidation rates have consequently been observed in the redoxcline where deep water enriched in $CH_4$ is mixed with oxic water (Jakobs et al., 2013). Due to the special characteristics of deep water areas isolated from the atmosphere, and with transitions between oxic and anoxic conditions, the Baltic Sea is a unique and suitable system for studying key processes in $CH_4$ cycling, in particular for investigating different oxidation pathways.

Surface water $CH_4$ concentrations in the open Baltic Sea are typically about 3.5–5 nM – only slightly oversaturated compared to the atmosphere (Gülzow et al., 2013). In contrast, in shallow near-shore areas, observations indicate a very different situation, with $CH_4$ concentrations ranging from 10 to 500 nM (Humborg et al., 2019; Myllykangas et al., 2020; Lundevall-Zara et al., 2021; Roth et al., 2022) with large temporal and spatial variations on small scales (e.g., Roth et al., 2022). Methane emissions to the atmosphere depend on the degree of oversaturation in the surface water, but also on wind speed and temperature (e.g., Wanninkhof, 2014). Estimated $CH_4$ emissions from different near-shore sites in the Baltic Sea display a large range due to substantial variations in the parameters that control gas transfer across the air-sea interface (Humborg et al.,

2019; Lundevall-Zara et al., 2021; Asplund et al., 2022; Roth et al., 2022; 2023). Short-term and small-scale variations cause considerable challenges for empirical estimates of fluxes over larger scales and longer periods in time.

Different processes in the $CH_4$ cycling do, however, produce certain "fingerprints" on the isotopic composition, similar to how the relative contributions of different atmospheric $CH_4$ sources determine long-term trends of $\delta^{13}C_{CH4a}$ (Lan et al., 2021). This can be helpful when assessing process rates. Observations in the Baltic Sea show a pronounced $^{13}$C-$CH_4$ enrichment in the redoxcline (Schmale et al., 2012; 2016; Jakobs et al., 2013; 2014; Gülzow et al., 2014), which is the result of a preferential oxidation of the lighter isotope. Similarly, $CH_4$ emissions to the atmosphere can produce a $^{13}$C-$CH_4$ enrichment in the surface water because of a preferential outgassing of the lighter isotope (Knox et al., 1992). The isotopic composition of $CH_4$ produced in sediments depends on the processes involved, i.e., $CO_2$ reduction or acetate fermentation (Reeburgh, 2007; see also Sect. 2.3.5), but can then be modified by oxidation processes in the pore water (Chuang et al., 2019).

Models can be useful for identifying limiting processes and constraining budgets even though not all rates are well known, through sensitivity experiments on process rates and parameterizations, as well as on the influence of changes in forcing of the system. Methane cycling has previously been investigated in both lake (e.g., Lopes et al., 2011; Greene et al., 2014; Tan et al., 2015; Stepanenko et al., 2016; Bayer et al., 2019) and ocean (e.g., Nihous and Masutani, 2006; Wåhlström and Meier, 2014; Malakhova and Golubeva, 2022) modeling studies. In the present study, $CH_4$ cycling and dynamics in the Baltic Sea are introduced into the coupled physical-biogeochemical Baltic Sea long-term and large-scale eutrophication model (BALTSEM), by expanding with state variables for both $^{12}$C-$CH_4$ and $^{13}$C-$CH_4$ concentrations (see Sect. 2.2). BALTSEM has previously been used in a similar approach where stable isotopes of dissolved inorganic carbon as well as dissolved and particulate organic carbon were included in the model in order to investigate constraints on process rates (Gustafsson et al., 2015).

Benthic $CH_4$ release and the isotopic composition of $CH_4$ produced in the sediments are not well known except for a few specific sites where *in-situ* measurements have been acquired. This means that the model is, at this point, somewhat poorly constrained. The main objective of this study is to use the model in concert with observed water column $CH_4$ concentrations and isotopic compositions to 1. identify and roughly quantify key $CH_4$ fluxes, 2. set up a preliminary $CH_4$ budget for the Baltic Proper (where measured profiles of $CH_4$ concentration and isotopic composition are available), and 3. perform sensitivity experiments on $CH_4$ concentration and isotopic composition depending on transport and transformation processes.

The motivation for implementing the $CH_4$ modeling on a large scale – with considerable spatial differences in terms of e.g., water and sediment properties as well as production, respiration, and sedimentation patterns – was utilizing the application of an already well-established model. BALTSEM has been described and validated in many publications, and it has been demonstrated that both physical (e.g., salinity, temperature, vertical mixing, lateral exchange, air-sea exchange) and biogeochemical (e.g., carbon and nutrient cycling and oxygen production/consumption) processes are largely described satisfactory (Gustafsson et al., 2012; 2017; Savchuk et al., 2012).

## 2 Material and methods

### 2.1 Area description

The Baltic Sea is a semi-enclosed brackish sea, connected to the North Sea via the shallow and narrow Danish straits. The system is characterized by a pronounced horizontal salinity gradient – going from the almost oceanic entrance area to the low-saline northernmost sub-basin – as well as a permanent salt dominated stratification, restricted water exchange with the North Sea, and long residence times (e.g., Stigebrandt and Gustafsson, 2003). As a result of strong stratification and long residence times, the central Baltic Sea is naturally susceptible to deep water de-oxygenation. Massively increased nitrogen (N) and

phosphorus (P) loads from the early 1950s to the mid-1980s has caused a large expansion of de-oxygenated deep water areas (e.g., Gustafsson et al., 2012). The loads have declined substantially from the peak values in the 1980's (e.g., Kuliński et al., 2022), although oxygen conditions have not yet improved in the central Baltic Sea (Hansson and Viktorsson, 2023).

### 2.2 The model

BALTSEM is a horizontally averaged, but vertically resolved process-oriented model that couples hydrodynamic and

biogeochemical modules in time-dependent, numerical simulations. In the model, the Baltic Sea is divided into thirteen coupled sub-basins (Figure S1), with geometric characteristics as summarized in Table S1. The hydrodynamic module has been described in detail by Gustafsson (2000; 2003), whereas the biogeochemical module has been described in detail by Savchuk (2002). Below, the two modules are qualitatively recapped (Sect. 2.2.1-2.2.2).

In this study, a new expanded version of the model, BALTSEM-CH4 v1.0, with state variables representing both $^{12}C$-$CH_4$ and

$^{13}C$-$CH_4$ is presented for the first time. Figure 1 illustrates processes involved in $CH_4$ cycling that are included in the model. This study focuses on the modeling of stable $CH_4$ isotopes: The $CH_4$ sources (i.e., river load and sediment release), boundary conditions (i.e., atmospheric $CH_4$ and $CH_4$ at the open ocean boundary), transport and transformation processes (i.e., $CH_4$ oxidation and air-sea exchange), as well as the isotopic fingerprints associated with these processes are described in Sect. 2.3. The model parameterizations for both hydrodynamic and biogeochemical processes (prior to the inclusion of $CH_4$) have been

described in detail in earlier publications (e.g., Gustafsson 2000; 2003; Gustafsson et al., 2012; 2014; 2017; Savchuk, 2002; Savchuk et al., 2012); this will not be repeated here. A list of all state variables in the model is included in Appendix A (Table A1).

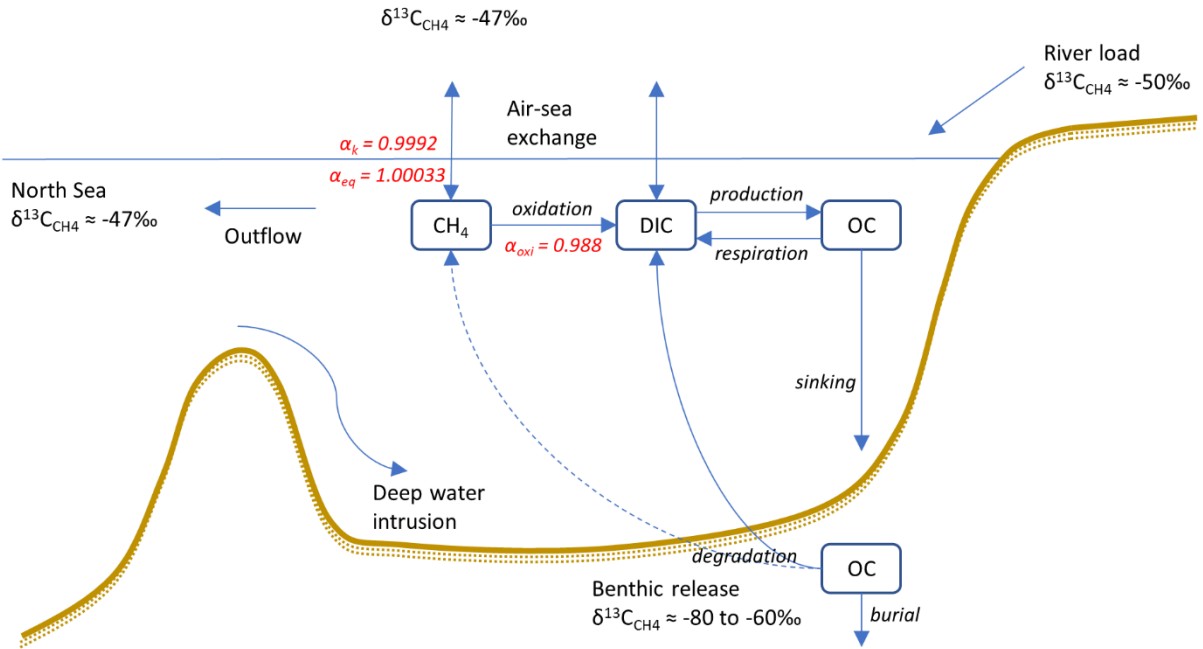

 **Figure 1: Conceptual sketch illustrating the processes involved in CH₄ cycling, including $\delta^{13}C_{CH4}$ values of end-members as well as $\alpha$ values of transformation processes (see Sect. 2.3). The benthic release (dashed arrow) is not explicitly modeled, instead a preset fitted value is used (see Sect. 2.3.5).**

### 2.2.1 Hydrodynamic module

The vertical stratification in each sub-basin is resolved by a variable number of horizontally homogenous layers. The numbers of layers in the respective sub-basins increase over time because of both inflows from adjacent basins and instances of pycnocline retreat, as described below, but are kept below maximum values by mixing of the two layers that require the least amount of energy to be merged (Gustafsson, 2000).

Flow dynamics through the straits that connect different sub-basins depends on the width of the strait compared to the internal
Rossby radius, determining whether or not earth rotation influence the water exchange. In general, lateral exchange between sub-basins is forced by barotropic pressure gradients across the straits that depend on sea level difference and wind set-up, as well as baroclinic pressure gradients caused by differences in stratification. In narrow straits, the water flow is influenced by frictional resistance and dynamical contraction due to the Bernoulli effect, while the transport through wider straits is further controlled by earth rotation effects (Gustafsson, 2000; 2003).

Dynamics of the mixed surface layer in each sub-basin is forced by wind stress and buoyancy fluxes, but also depends on earth rotation, following Stigebrandt (1985). The pycnocline is eroded whenever the buoyancy flux is negative (e.g., if surface water

density increases because of net evaporation, or by cooling when the water temperature is above the temperature for maximum density), or when the buoyancy flux is positive but the power generated by wind stress is sufficient to do work against the buoyancy forces. Pycnocline erosion means that the mixed surface layer becomes thicker and denser as a result of deep-water entrainment into the surface layer. If the power is not sufficient, the turbulent mixing becomes limited either by earth rotation or by buoyancy fluxes, leading to a pycnocline retreat and the formation of a new and shallower mixed surface layer. The thickness of the new surface layer will be determined either by the Ekman or Monin-Obukov length-scale – whichever is shorter (Stigebrandt, 1985).

Entrainment flows are further modified by the presence of sea-ice (Gustafsson, 2003). Ice dynamics is based on a sea-ice model by Björk (1997), but adapted to the Baltic Sea following Nohr et al. (2009). Calculations for heating/cooling and evaporation at the sea, ice or snow surface follow Björk (1997). About half of the incoming short-wave radiation is absorbed at the surface while the remaining fraction attenuates exponentially using constant attenuation factors for water, ice and snow, respectively. Turbulent vertical diffusion in deeper layers below the mixed surface layer is parameterized as a function of stratification and mixing wind (Stigebrandt, 1987; Axell, 1998), representing the energy inputs from inertial currents and breaking internal waves. The model further includes dense gravity currents (i.e., deep-water inflows along the seafloor), where entrainment of surrounding deep water into the gravity currents depends on bottom slope and friction as well as density difference between the gravity current and the surrounding water (Stigebrandt, 1987). Entrainment of surrounding water into gravity currents has the effect that the volume flow increases while at the same time density decreases, influencing at what depth the gravity current will be interleaved, i.e., the depth of neutral buoyancy. Deep-water inflows cause an uplift of the entire water column above the intrusion depth.

### 2.2.2 Biogeochemical module

Biogeochemical processes are calculated using a nutrient-phytoplankton-zooplankton-detritus model setup that closely follows Savchuk (2002), but that has been expanded with state variables representing e.g., dissolved organic compounds and the inorganic carbon system (Gustafsson et al., 2014).

The biogeochemical module includes pelagic state variables for oxygen ($O_2$), hydrogen sulfide ($H_2S$), total alkalinity, dissolved inorganic carbon, nitrate + nitrite, ammonium, phosphate, dissolved silica, labile and refractory fractions of dissolved organic carbon (C), nitrogen (N), and phosphorus (P), particulate organic C, N, P, and silicon (Si), three functional groups of phytoplankton (representing diatoms, 'summer species', and diazotrophic cyanobacteria), and one bulk state variable for heterotrophs that represents zooplankton and other organisms that consume and mineralize phytoplankton and detrital matter. All pelagic state variables are subject to transport processes (vertical mixing and horizontal advection) as well as various biological and chemical transformation processes; source and sink terms for each state variable are computed in all water layers in each sub-basin. BALTSEM further includes sediment pools of C, N, P, and Si that are subject to mineralization and burial. The pelagic and benthic realms are coupled by sedimentation of organic matter and sediment-water exchange of

dissolved inorganic compounds. Oxygen, $CO_2$, and $CH_4$ are exchanged at the air-sea boundary depending on solubilities, wind speed, and gradients between sea surface and air of the respective gases.

Phytoplankton growth depends on water temperature and is further limited by light and nutrient availability (Savchuk, 2002). Light penetration in water in the biogeochemical module is calculated as a function of the biogeochemical state. The phytoplankton groups assimilate dissolved inorganic C, N, and P according to fixed Redfield ratios while at the same time producing oxygen, but also take up an excess of dissolved inorganic carbon which is transformed into dissolved organic carbon, representing extracellular production (Gustafsson et al., 2014). The cyanobacteria group is able to fix atmospheric N when ammonium and nitrate become limiting. The diatom group is the only phytoplankton group that requires dissolved silica. Loss terms for phytoplankton include natural mortality, grazing by zooplankton, and sinking. Dead phytoplankton are converted into detrital C, N, P, and Si according to their elemental stoichiometry.

Heterotroph/zooplankton growth depends on grazing rate which is regulated by water temperature and food concentration (phytoplankton and detritus) as well as the respective availability of different food sources (Savchuk, 2002). Grazing is in addition strongly inhibited at low oxygen concentrations. Fractions of each food source that are not digested are instead assigned to detritus pools in accordance with stoichiometry of the food sources. Zooplankton have elemental stoichiometry that differ from their food sources; growth thus becomes limited by the element in relative shortage, while carbon and nutrients in excess compared to zooplankton stoichiometry are excreted. Zooplankton biomass decreases by natural mortality and excretion; dead zooplankton are converted into detrital C, N, and P according to elemental stoichiometry.

Phytoplankton and detritus sink through the water column; phytoplankton that are not lost by grazing or natural mortality in the water column settle on the seafloor where their constituents are assigned to sediment pools of C, N, P, and Si according to elemental composition. Temperature dependent leaching converts a fraction of the detritus into dissolved organic C, N, and P, as well as dissolved silica in the water column, while the remainder is either consumed by zooplankton in the water column or settles on the seafloor where it is assigned to the respective sediment pools. Organic carbon and nutrients in the water column are mineralized either by means of zooplankton respiration (dissolved inorganic carbon) and excretion (ammonium and phosphate) or by temperature dependent oxidation of dissolved organic compounds; these processes also consume oxygen. Nitrification converts ammonium into nitrate while consuming oxygen. Heterotrophic and chemolithoautotrophic denitrification processes represent loss terms for nitrate. In the absence of both oxygen and nitrate, organic matter is instead oxidized by sulfate, which also leads to hydrogen sulfide production. Sulfide can be oxidized by either oxygen or nitrate (i.e., chemolithoautotrophic denitrification); sulfide oxidation thus represents loss terms for either oxygen or nitrate.

The sediment compartment in each sub-basin can be described as a series of horizontal terraces with a resolution of one terrace per one meter water depth; the area of each terrace is a function of the hypsographic curve for the respective sub-basins. Sediment state variables are not vertically resolved on the individual terraces, but instead formulated as pools of bioavailable C, N, P, and Si that have been deposited on the different terraces – representing the "active" (i.e., not permanently sequestered) top layer of sediments (Savchuk et al., 2012). The carbon and nutrients in phytoplankton and detritus that settles on the terraces are added to the respective sediment pools. A fraction of the sediment pools is permanently sequestered and thus removed

from the biogeochemical cycling, while the remaining fraction undergoes temperature dependent mineralization into inorganic carbon and nutrients that can again be released to the water column.

Nutrient cycling and release from the sediments is strongly coupled to oxygen concentration in the overlying water. During oxic conditions, mineralized N is released in the form of nitrate, but an oxygen dependent fraction of the nitrate is lost by denitrification. A fraction of the mineralized P is retained in the sediments during oxic conditions, representing phosphate bound to e.g., iron oxides. P retention capacity is further regulated by salinity, representing a proxy for both sulfate concentration and iron availability (Savchuk et al., 2012). During anoxic conditions in the overlying water, mineralized N is

released in the form of ammonium. At the same time, mineralized P cannot be retained in the sediments during anoxic conditions; instead, previously sequestered phosphate is released to the water column, representing reduction of metal oxides that are thus unable to bind phosphate. During oxic conditions, sediment mineralization consumes oxygen in the overlying water; during anoxic conditions, the sediments release hydrogen sulfide to the overlying water, representing sulfate reduction.

### 2.2.3 Model forcing, boundary conditions and initial conditions

The meteorological forcing includes three-hourly wind data, air temperature, cloudiness, air pressure, and precipitation. Model forcing for the hydrodynamic module also includes observed daily mean sea level in the Kattegat as well as monthly mean river runoff to each sub-basin. Further, the model forcing includes monthly mean loads of inorganic and organic carbon and nutrients, and alkalinity from land (point sources and river loads) and atmosphere. Daily profiles of salinity and temperature (i.e., stratification), as well as concentrations of all biogeochemical state variables (Table A1) define the conditions at the open

boundary between the Northern Kattegat (sub-basin 1; Figure S1) and the Skagerrak (open ocean). Monthly mean atmospheric partial pressures of $CO_2$ and $CH_4$ comprise the atmospheric boundary conditions for the respective gases. The model forcing is further detailed in Appendix B.

An initial model run over the period 1970-2000 started with initial profiles for the different state variables based on observations when possible or else fitted values. The initial model run was then used as a spin-up for a series of model runs

covering the period 2001-2020 that are performed to examine the sensitivity of e.g., $CH_4$ concentration and isotopic composition depending on process parameterizations (Sect. 4.1).

### 2.3 Methane modeling

### 2.3.1 Isotopic fractionation

Isotope values of $CH_4$ are expressed in $\delta^{13}C$ units (‰) relative to the Vienna Peedee Belemnite (VPDB) standard (Hoffman and Rasmussen, 2022):

$$\delta^{13}C = \left(\frac{R_{sample}}{R_{std}} - 1\right) * 1000,$$ (1)

Here, $R_{sample}$ and $R_{std}$ represent the $^{13}C/^{12}C$ ratios of a sample and the PDB standard, respectively.

Isotopic fractionation $\alpha$ during different processes (e.g., oxidation, air-sea exchange) in the $CH_4$ cycling can be expressed as:

$$\alpha = \frac{R_A}{R_B}, \tag{2}$$

Here, $R_A$ and $R_B$ represent $^{13}C/^{12}C$ ratios of compounds $A$ and $B$.

Fractionation can also be expressed in $\delta^{13}C$ units using Eq. 1 and 2:

$$\alpha = \left(\frac{\delta_A}{1000} + 1\right) / \left(\frac{\delta_B}{1000} + 1\right), \tag{3}$$

Alternatively, fractionation is often expressed as $\varepsilon$ values (Zeebe and Wolf-Gladrow, 2001):

$$\varepsilon = \delta_A - \delta_B \approx (\alpha - 1) * 1000, \tag{4}$$

In the model description below, both $\alpha$ and $\varepsilon$ values are used to describe fractionation during different processes.

**2.3.2 Air-sea exchange**

The $CH_4$ flux ($F_{CH4}$) between water and air is calculated according to:

$$F_{CH4} = k(CH_{4eq} - CH_{4w}), \tag{5a}$$
$$CH_{4eq} = K_0 pCH_{4a}, \tag{5b}$$


where $k$ (m s$^{-1}$) is the transfer velocity, $CH_{4eq}$ the equilibrium concentration with the atmosphere, $K_0$ (nM atm$^{-1}$) the $CH_4$ solubility, $pCH_{4a}$ (atm) the partial pressure of $CH_4$ in air, and $CH_{4w}$ (nM) the $CH_4$ concentration in surface water.

The solubility is calculated as a dimensionless Bunsen coefficient ($\beta$) according to Wiesenburg and Guinasso (1979):

$$\ln\beta = A_1 + A_2(100/TK) + A_3\ln(TK/100) + S(B_1 + B_2(TK/100) + B_3(TK/100)^2), \tag{6}$$

Here, $A_1$, $A_2$, $A_3$, $B_1$, $B_2$, and $B_3$ are constants and $TK$ is temperature (K).

$\beta$ is then converted to $K_0$ (nM atm$^{-1}$) according to:

$$K_0 = \frac{P_S \beta}{R T_S} * 10^6, \tag{7}$$

where $P_S = 101325$ Pa atm$^{-1}$ represents a unit conversion from Pa to atm, $R = 8.314$ m$^3$ Pa K$^{-1}$ mol$^{-1}$ is the molar gas constant, and $T_S = 273.15$ K is the standard temperature.

The transfer velocity $k$ is calculated according to Wanninkhof (2014) and converted from cm hour$^{-1}$ to m s$^{-1}$:

$$k = 0.251 U_{10}^2 \sqrt{\frac{660}{Sc}} * \frac{0.01}{3600}, \tag{8}$$

Here, $U_{10}$ (m s$^{-1}$) is the wind speed at 10 m height and $Sc$ is the Schmidt number for $CH_4$ (Wanninkhof, 2014):

$$Sc = A + BT + CT^2 + DT^3 + ET^4, \tag{9}$$

Here, $A$, $B$, $C$, $D$, and $E$ are constants and $T$ is temperature (°C).

The atmospheric $CH_4$ level has increased from around 800 ppb to almost 1900 ppb over the last two centuries (see Fig. S2). In the different model runs, the atmospheric $CH_4$ levels according to the RCP 4.5 scenario were used (Fig. S2). The mixing ratio is expressed as mole fraction of dry air (ppb) and thus identical to the $CH_4$ partial pressure, $pCH_{4a}$ (natm).

*Fractionation during gas transfer and dissolution*

The fractionation of a gas during transfer between air and water depends on two fractionation processes – gas dissolution and molecular gas transfer. The fractionation $\alpha_{eq}$ during dissolution of $CH_4$ in water is defined as (Knox et al., 1992):

$$\alpha_{eq} = \frac{R_{CH4eq(d)}}{R_{CH4eq(g)}}, \tag{10}$$

Here, $R_{CH4eq(d)}$ and $R_{CH4eq(g)}$ represent the ratios of the heavy and light $CH_4$ isotopes between the equilibrium concentrations of $CH_4$ in dissolved (d) and gas phase (g), respectively. Experiments by Fuex (1980) indicate that the heavy $CH_4$ isotope is more soluble than the lighter isotope (although the lighter isotope initially dissolves faster), with a fractionation during dissolution amounting to approximately $\alpha_{eq} = 1.00033$.

A difference in the molecular transfer rates of heavy and light $CH_4$ isotopes, result in further fractionation defined as (Knox et al., 1992):

$$\alpha_k = \frac{k_{13CH4}}{k_{12CH4}}, \tag{11}$$

Here, $k_{13CH4}$ and $k_{12CH4}$ represent the transfer rates of the heavy and light isotope, respectively. Experiments by Knox et al. (1992) indicate a preferential exchange of the light isotope, with a fractionation during gas transfer of approximately $\alpha_k = 0.9992$. Measurements from stagnant wooded swamps point to a reduced gas exchange but also a considerably more pronounced kinetic fractionation in waters with insoluble organic surface films (Happell et al., 1995). Surface films are, however, not taken into account in BALTSEM-CH$_4$ v1.0.

The $^{13}$C-CH$_4$ flux between water and air, $F_{13CH4}$, is calculated based on Holmes et al. (2000):

$$F_{13CH4} = k\alpha_k \left( K_0 pCH_{4a} R_{atm} \alpha_{eq} - \left[ ^{13}CH_{4w} \right] \right), \tag{12}$$

Here, $R_{atm}$ is the $^{13}$C/$^{12}$C ratio of atmospheric CH$_4$, and [$^{13}$CH$_{4w}$] is the surface water concentration of $^{13}$C-CH$_4$.

In the model runs, the atmospheric $\delta^{13}C_{CH4}$ is set to a constant -47‰.

### 2.3.3 River loads

Measurements in Swedish low-order streams (Strahler stream order 1-4) indicate a median CH$_4$ concentration of approximately 6.7 µg C L$^{-1}$ – corresponding to 560 nM – but with substantial variations between individual streams (Wallin et al., 2018). As opposed to CO$_2$ concentrations that generally declined with increasing stream order, there was no such clear relation between stream order and median CH$_4$ concentration, although the lowest median concentration (3.6 µg C L$^{-1}$, corresponding to 300 nM) was reported for the largest streams (Wallin et al., 2018).

CH$_4$ produced in freshwater sediments and wetlands is presumably mainly resulting from acetate fermentation (see Sect. 2.3.5), with isotope values in a typical range -65‰ to -50‰ (Whiticar et al., 1986; Quay et al., 1988). However, both CH$_4$ oxidation and outgassing cause a $^{13}$C enrichment in the residual CH$_4$ pool. This means that an increasing isotope value is expected as outgassing and oxidation processes gradually modulate both CH$_4$ concentrations and isotopic composition in streams and rivers along their routes towards the sea. Measurements in a subtropical river network in Australia indicate surface water $\delta^{13}C_{CH4}$ values in a range -57‰ to -47‰ (Atkins et al., 2017), i.e., values close to, or lower than the atmospheric $\delta^{13}C_{CH4}$ (see Sect. 2.3.2). Similarly, measurements in an urbanized river system in Scotland indicate $\delta^{13}C_{CH4}$ values in a range -60‰ to -47‰ (Gu et al., 2021).

As a fist approximation, it will be assumed that the riverine CH$_4$ concentration ($CH_{4riv}$) is 100 nM and that $\delta^{13}C_{CH4}$ = -50‰ in rivers entering the Baltic Sea.

### 2.3.4 Inflows from the North Sea

Methane concentrations in open North Sea surface waters are highly heterogeneous, but generally above the solubility equilibrium with the atmosphere. Observations indicate a range 3 - 30 nM (Bange et al., 1994; Rehder et al., 1998; Osudar et al., 2015). This heterogeneity has been suggested to partly be a result of the westward transport of surface waters originating from the Kattegat and Skagerrak (Rehder et al., 1998). Closer to the coasts where both rivers and coastal sediments can be

significant regional sources of $CH_4$, concentrations are usually considerably higher (Scranton and McShane, 1991; Rehder et
al., 1998; Upstill-Goddard et al., 2000; Grunwald et al., 2009; Osudar et al., 2015), but large fractions appear to be removed
within estuaries before reaching the open sea (Upstill-Goddard et al., 2000; Grunwald et al., 2009). Measurements from the
southern central North Sea indicate concentrations close to (but higher than) the equilibrium (Scranton and McShane, 1991;
Bange et al., 1994; Rehder et al., 1998).

As a fist approximation, it will be assumed that the $CH_4$ concentration is 5 nM and that $\delta^{13}C_{CH4} = -47‰$ in North Sea water
entering the Baltic Sea.

### 2.3.5 Benthic release

*Methanogenesis*

There are two primary methanogenic pathways for biologically mediated $CH_4$ production – $CO_2$ reduction and acetate
fermentation (Reeburgh, 2007):

$$CO_2 + 4H_2 \rightarrow CH_4 + 2H_2O, \tag{13}$$
$$CH_3COOH \rightarrow CH_4 + CO_2, \tag{14}$$

$CO_2$ reduction is dominant in the sulfate depleted zone of marine sediments, whereas acetate fermentation is dominant in
freshwater sediments. Both pathways may nevertheless occur in both marine and limnic environments (Whiticar et al., 1986).
Methanogenesis in marine environments is assumed to predominantly occur in anoxic sediments, whereas the presence of
oxygen and/or sulfate generally prevents large-scale methanogenesis in the water column. In anoxic sediments, sulfate can be
used as an oxidant during mineralization of organic matter or consumed by sulfate mediated oxidation of $CH_4$. The sediment
depth of sulfate depletion and the main zone of methanogenesis depend strongly on location and sedimentation rate.
Measurements in the Baltic Sea area indicate sulfate depletion depths in a range of centimeters to meters (Jørgensen et al.,
1990; Slomp et al., 2013; Myllykangas et al., 2020).

The default sediment source is set to 50 µmol m$^{-2}$ d$^{-1}$, which is a fitted value that produces deep water $CH_4$ observations
reasonably well. The impact from the sediment source is further explored in different sensitivity experiments (Sect. 4.1).

*Fractionation during methanogenesis*

Isotope values of $CH_4$ from biogenic sources are typically in a range -110‰ to -60‰ (Whiticar et al., 1986), depending on
methanogenic substrate and mechanisms (i.e., $CO_2$ reduction vs. acetate fermentation). The fractionation during $CO_2$ reduction
is typically $\varepsilon > 95‰$, while the fractionation during acetate fermentation is in a range $\varepsilon \sim 40\text{-}60‰$ (Whiticar, 1999).

Measurements from anoxic deep water in the central Baltic Sea show isotope values of -84‰ and -71‰ in the Gotland and
Landsort Deeps, respectively (Jakobs et al., 2013). Deep water in the Bornholm basin shows isotope values of approximately
-70‰, and deep water in the Arkona basin show isotope values in a range -69‰ to -63‰ (Gülzow et al., 2014). Measurements

by Roth et al. (2022) indicate a value of approximately -67‰ for the sediment source in shallow areas with oxic conditions in the water column. Furthermore, measurements by Egger et al. (2017) indicate surface sediment pore water $\delta^{13}$C-CH$_4$ values of approximately -80‰ in the Landsort Deep (451 mbss), -70‰ in the Bornholm Deep (87 mbss), and -60‰ in the Little Belt
(37 mbss).

As a first approximation, the sediment CH$_4$ source in the model is assumed to have a $\delta^{13}$C-CH$_4$ value of -80‰ or -60‰ in sediments underlying anoxic or oxic water, respectively. This value will then be adjusted in different sensitivity experiments (Sect. 4.1).

### 2.3.6 Methane oxidation in the water column

*Methane oxidation by oxygen (MOX)*

Aerobic CH$_4$ consumption by methanotrophic processes consume CH$_4$ and oxygen while producing CO$_2$ and water:

$$CH_4 + 2O_2 \rightarrow CO_2 + 2H_2O, \tag{15}$$

The oxidation rate, $W_{CH4\_O2}$ (nM day$^{-1}$) is parameterized as Monod functions of both CH$_4$ (nM) and O$_2$ (µM) concentrations (Van Bodegom et al., 2001; Greene et al., 2014):

$$W_{CH4\_O2} = v_{WCH4\_O2} \left( \frac{[CH_4]}{h_{CH4}+[CH_4]} \right) \left( \frac{[O_2]}{h_{O2}+[O_2]} \right), \tag{16}$$

Here, $v_{WCH4\_O2}$ (nM d$^{-1}$) is the potential maximum aerobic oxidation rate, *[CH$_4$]* (nM) and *[O$_2$]* (µM) are CH$_4$ and O$_2$ concentrations, whereas $h_{CH4}$ (nM) and $h_{O2}$ (µM) are "half saturation" concentrations for CH$_4$ and O$_2$, respectively.

Aerobic oxidation of CH$_4$ is also included as an O$_2$ sink term in the model, consuming 2 mol O$_2$ for each mol of consumed CH$_4$ (Eq. 15). Furthermore, both aerobic and anaerobic CH$_4$ oxidation are included as sources of dissolved inorganic carbon in the model, producing 1 mol of dissolved inorganic carbon for each mol of consumed CH$_4$ (Eq. 15 and 17, respectively).

Observations from the Gotland and Landsort deeps in the Baltic Proper indicate oxidation rates in a range 0.1-4 nM d$^{-1}$ depending on location and season (Schmale et al., 2012; 2016; Jakobs et al., 2014). The parameters in Eq. 16 are set to default values of $v_{WCH4\_O2}$ = 8 nM d$^{-1}$, $h_{CH4}$ = 60 nM, and $h_{O2}$ = 100 µM. These are fitted values that produce CH$_4$ concentrations and oxidation rates that fairly well reproduce observations (see Sect. 3.1). In Sect. 4.1, the influence of modified values of $v_{WCH4\_O2}$, $h_{CH4}$, and $h_{O2}$ will be addressed in different sensitivity experiments.


*Anaerobic oxidation of CH$_4$ by sulfate (AOM)*

AOM is typically assumed to be mediated by sulfate, although other oxidants such as nitrate/nitrite and also iron and manganese oxides could be used as well (Myllykangas et al., 2020). The stoichiometry for sulfate mediated AOM can be written (Hoehler et al., 1994):


$$CH_4 + SO_4^{2-} \rightarrow HS^- + HCO_3^- + H_2O, \tag{17}$$

The ratios of sulfate and chlorine concentrations in the Baltic Sea are close to the oceanic ratio (Kremling, 1972), which means that the sulfate concentration $[SO_4^{2-}]$ in the Baltic Sea can be approximated as


$$[SO_4^{2-}] \approx [SO_4^{2-}]_{oc} \frac{S}{35}, \tag{18}$$

Here, $S$ is the salinity and $[SO_4^{2-}]_{oc} = 0.0282$ mol kg$^{-1}$ is the sulfate concentration in sea water ($S = 35$) (Dickson et al., 2007). Thus, sulfate concentrations in the Baltic Sea are orders of magnitude higher than $CH_4$ concentrations. For that reason, $CH_4$

oxidation by sulfate, $W_{CH4\_SO4}$ (nM day$^{-1}$) is parameterized as a function of $CH_4$, whereas the sulfate concentration is assumed not to be limiting in the water column:

$$W_{CH4\_SO4} = v_{WCH4\_SO4} \left( \frac{[CH_4]}{h_{CH4}+[CH_4]} \right), \tag{19}$$

Since sulfate is assumed not to be limiting, other potential oxidants during AOM are not accounted for in the model. Observations from the anoxic deep waters of the Gotland and Landsort deeps in the Baltic Proper indicate oxidation rates < 0.1 nM d$^{-1}$ (Jakobs et al., 2014). The maximum anaerobic oxidation rate is set to a default value of $v_{WCH4\_SO4} = 0.1$ (nM d$^{-1}$).

*Fractionation during CH$_4$ oxidation*

There is a preferential oxidation of $^{12}$C-CH$_4$ compared to the heavier $^{13}$C-CH$_4$, causing a fractionation during the process. The oxidation of $^{13}$C-CH$_4$ is thus computed according to:

$$W_{13CH4\_O2} = \alpha_{oxi} R_{CH4} W_{CH4\_O2}, \tag{20}$$

Here, $\alpha_{oxi}$ is the fractionation during CH$_4$ oxidation, $R_{CH4}$ is the $^{13}$C/$^{12}$C ratio of CH$_4$, and $W_{CH4\_O2}$ is the CH$_4$ oxidation rate (Eq. 16). By use of Eq. 16, Eq. 20 can be rewritten as:

$$W_{CH4\_O2} \approx \alpha_{oxi} v_{WCH4\_O2} \left( \frac{[^{13}CH_4]}{h_{CH4}+[CH_4]} \right) \left( \frac{[O_2]}{h_{O2}+[O_2]} \right), \tag{21}$$

Observations indicate a wide range of fractionation during $CH_4$ oxidation (e.g., $\varepsilon \sim$ 4-30‰, Whiticar (1999); $\varepsilon \sim$ 16-54‰, Chan et al. (2019b)). Based on observations from the central Baltic Sea by Jakobs et al. (2013), the default fractionation is set to 12‰, which corresponds to $\alpha_{oxi}$ = 0.988 in Eq. 20. The influence of fractionation during $CH_4$ oxidation is addressed by sensitivity experiments in Sect. 4.1.

## 3 Results

In this section, simulated $CH_4$ concentrations, isotopic compositions, and aerobic and anaerobic oxidation rates are presented for a 'standard' model run (Sect. 3.1). Simulated large-scale fluxes and a preliminary $CH_4$ budget are presented in Sect. 3.2.

### 3.1 Standard model run

The standard model run was performed over the period 2001-2020 after spin-up (see Sect. 2.2) with parameters as indicated in Table 1. These parameters (i.e., $CH_4$ oxidation rates and fractionation values, $CH_4$ sources from the sediments, rivers, and 450   the North Sea, as well as the isotopic compositions of these sources) are mostly fitted values, where the intention was to reasonably well reproduce existing observations of both $CH_4$ concentration and isotopic composition from the Gotland Sea. This simulation will then be used as a basis for the sensitivity experiments presented in Sect. 4.1. Simulated contour plots and time series for the period 2001-2020 are presented in Fig. 2-3. Furthermore, monthly mean profiles for years 2014 and 2015 are presented in Fig. 4-5 in order to illustrate seasonal dynamics in surface waters as well as the impact of a major deep water 455   inflow.

Table 1. Standard model settings. The values are own estimates/fitted values (see Sect. 2.3) except where noted.

| Parameter | Notation | Value | Unit |
|---|---|---|---|
| Potential maximum oxidation rate (MOX) | $v_{WCH4\_O2}$ | 8 | nM d$^{-1}$ |
| Potential maximum oxidation rate (AOM) | $v_{WCH4\_SO4}$ | 0.1 | nM d$^{-1}$ |
| Half saturation value, $CH_4$ oxidation | $h_{CH4}$ | 60 | nM |
| Half saturation value, $CH_4$ oxidation | $h_{O2}$ | 100 | µM |
| Fractionation, $CH_4$ oxidation | $\alpha_{oxi}$ | 0.988[1] | - |
| Sediment source, $CH_4$ flux | $r_{SED}$ | 50 | µmol m$^{-2}$ d$^{-1}$ |
| Sediment source, $\delta^{13}C$-$CH_4$, anoxic water | $\delta^{13}C\text{-}CH_{4sed}$ | -80 | ‰ |
| Sediment source, $\delta^{13}C$-$CH_4$, oxic water | $\delta^{13}C\text{-}CH_{4sed}$ | -60 | ‰ |
| Riverine $CH_4$ | $CH_{4riv}$ | 100 | nM |
| Riverine $\delta^{13}C$-$CH_4$ | $\delta^{13}C\text{-}CH_{4riv}$ | -50 | ‰ |
| North Sea $CH_4$ | $CH_{4NS}$ | 5 | nM |

| North Sea $\delta^{13}$C-CH$_4$ | $\delta^{13}C\text{-}CH_{4NS}$ | -47 | ‰ |
|---|---|---|---|

1. Jakobs et al. (2013)

Figure 2 illustrates the characteristic dynamics of the permanently salinity stratified Gotland Sea. The top of the halocline, which is typically located at around 60 m depth, isolates deeper waters from the atmosphere which means that $O_2$ can only be supplied via deep water inflows of oxic and comparatively high-saline water and through vertical turbulent diffusion. Stagnation periods with little or no advective $O_2$ supply to the deep may last for years, and since $O_2$ consumption by degradation processes exceeds the turbulent diffusive flux eventually conditions anoxic prevails. Stagnation periods are also characterized

by CH$_4$ accumulation because of a low anaerobic oxidation rate, and the $\delta^{13}$C-CH$_4$ in anoxic water is also close to the sediment source because of the marginal influence of anaerobic oxidation processes in the water column. Inflows of new deep water lead to an uplift of the water column above the intrusion depth, which is clearly seen in the simulated $O_2$ and CH$_4$ profiles in February to June of 2015 (Fig. 5, upper panel). Inflows furthermore cause a sharp decline in deep water CH$_4$ concentration (Fig. 2-3), primarily due to water exchange, but additionally because of high aerobic oxidation rates during periods when $O_2$

and CH$_4$ co-occur in the deep water until $O_2$ is again depleted (Fig. 5).

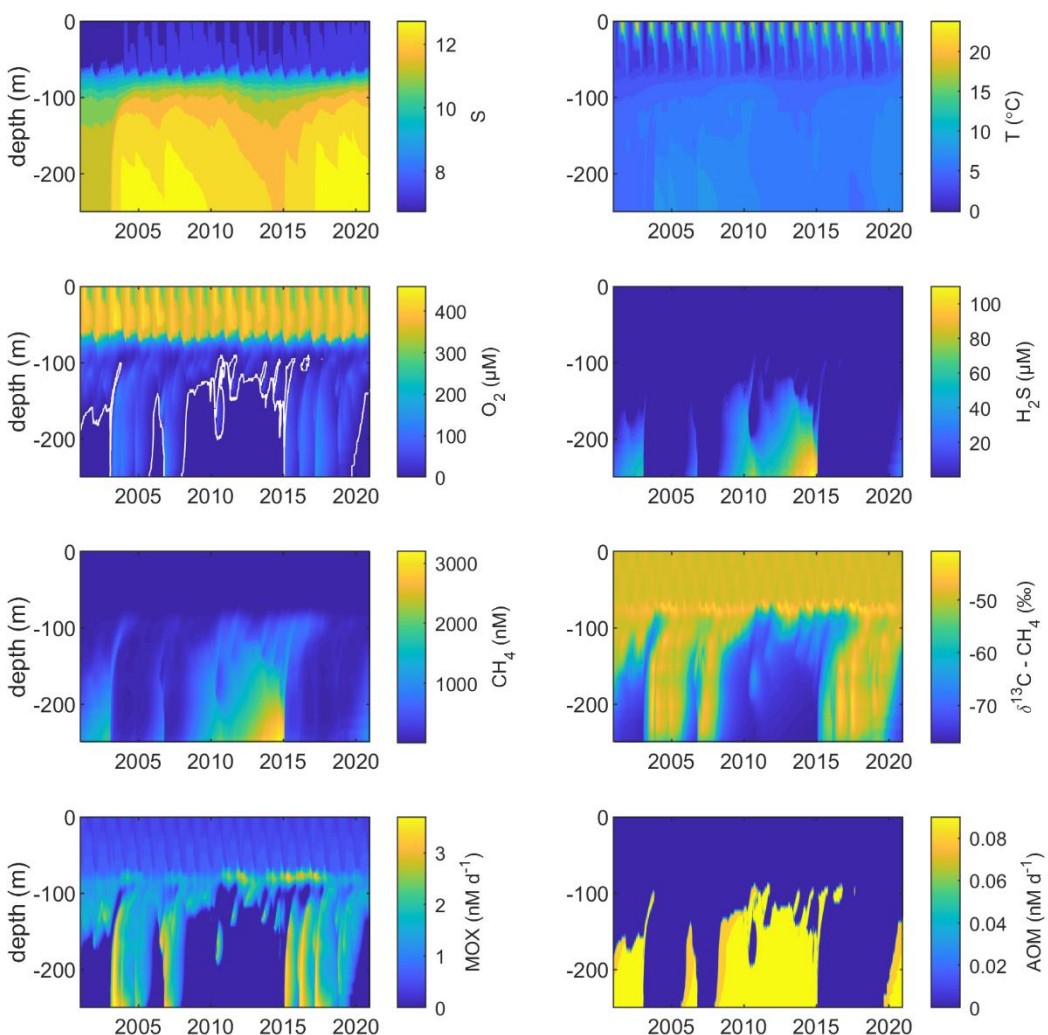

**Figure 2: Model output from the standard model run, showing simulated S, T (˚C), O₂ (µM), H₂S (µM), CH₄ (nM), δ¹³C-CH₄ (‰), MOX (nM d⁻¹), and AOM (nM d⁻¹) in the Gotland Sea sub-basin (cf. Fig. S1) over the period 2001-2020. The white line in the O₂ plot indicates the upper limit for anoxic deep water.**


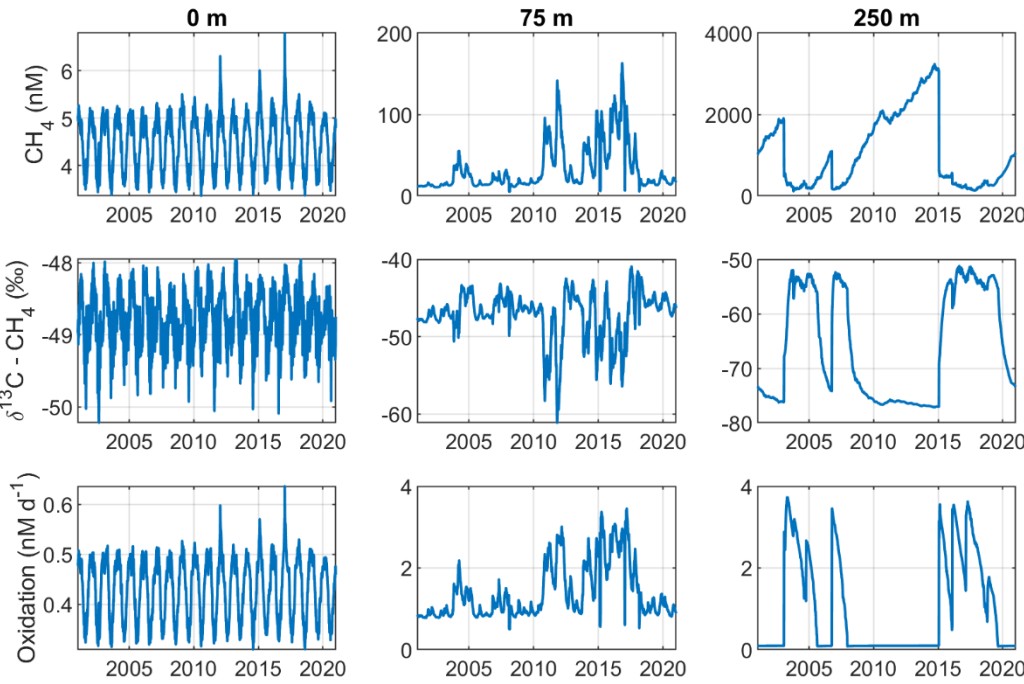

**Figure 3: Model output from the standard model run, showing simulated surface (0 m; left-hand panels), intermediate (75 m; middle panels), and deep water (250 m; right-hand panels) development of CH₄ (nM), CH₄ oxidation (MOX + AOM; nM d⁻¹), and δ¹³C-CH₄ (‰) in the Gotland Sea sub-basin (cf. Fig. S1) over the period 2001-2020.**

In surface waters above the top of the halocline, seasonal changes in temperature and thermal stratification largely influence other parameters (Fig. 4-5; see also Fig. S3-S4, supporting information). The increasing surface water temperature in spring and summer leads to decreasing $O_2$ and $CH_4$ solubility which in addition affects aerobic oxidation rates that depend on $O_2$ and $CH_4$ concentrations (Fig. S3-S4, supporting information; see also Eq. 16). This temperature dependence on oxidation rates also has an impact on the isotopic composition of $CH_4$ – the $\delta^{13}C$-$CH_4$ in water above the top of the halocline is strongly influenced by the seasonality of temperature stratification (Fig. S3-S4, supporting information). However, the variations of isotopic composition in surface waters are significantly smaller than the variations at depth where $\delta^{13}C$-$CH_4$ mainly depends on transitions between oxic and anoxic conditions (Fig. 4-5).

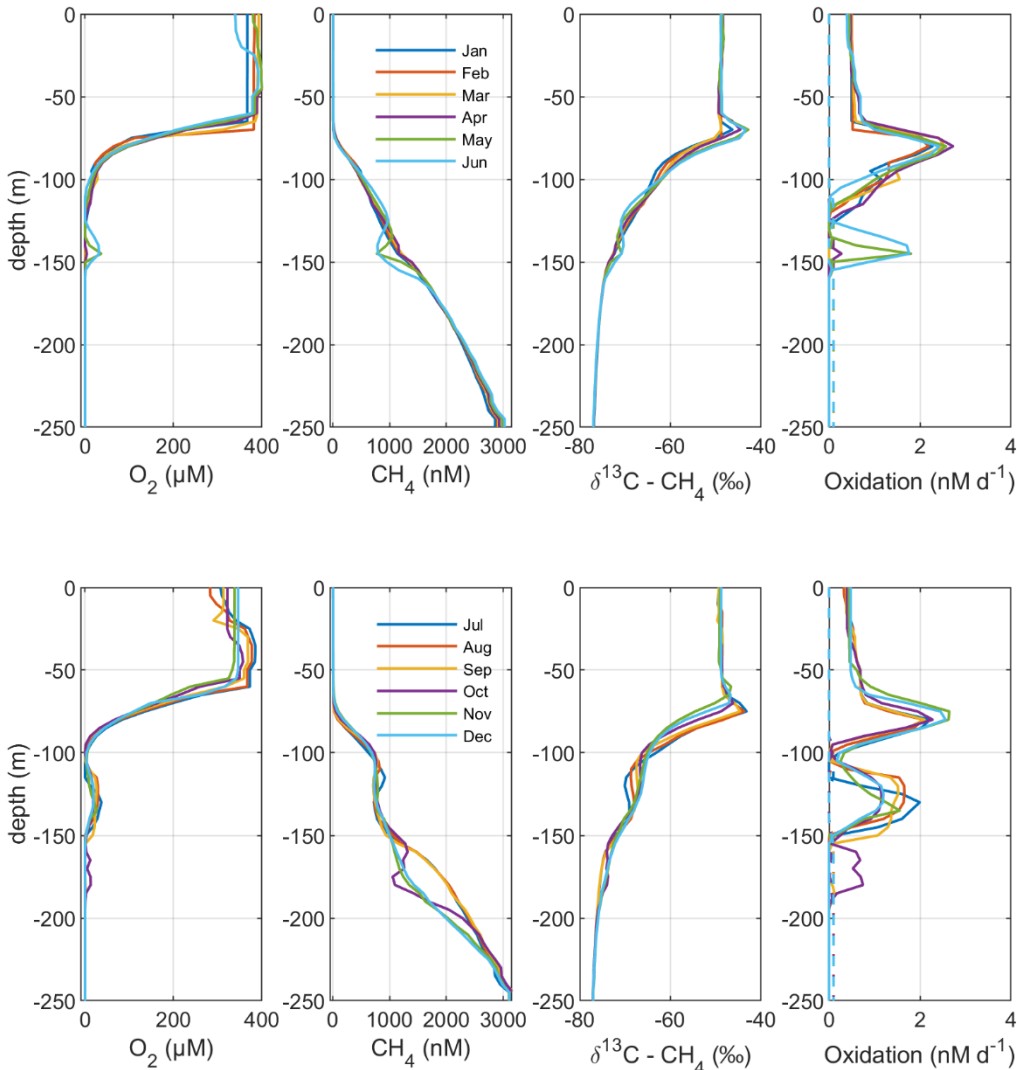

**Figure 4: Model output from the standard model run, showing simulated monthly mean profiles of CH₄ (nM), δ¹³C-CH₄ (‰), and oxidation rates (nM d⁻¹; MOX – full lines, AOM – dashed lines) in the Gotland Sea sub-basin (cf. Fig. S1) year 2014. The upper panels illustrate monthly mean profiles from January to June; the lower panels illustrate monthly mean profiles from July to December.**

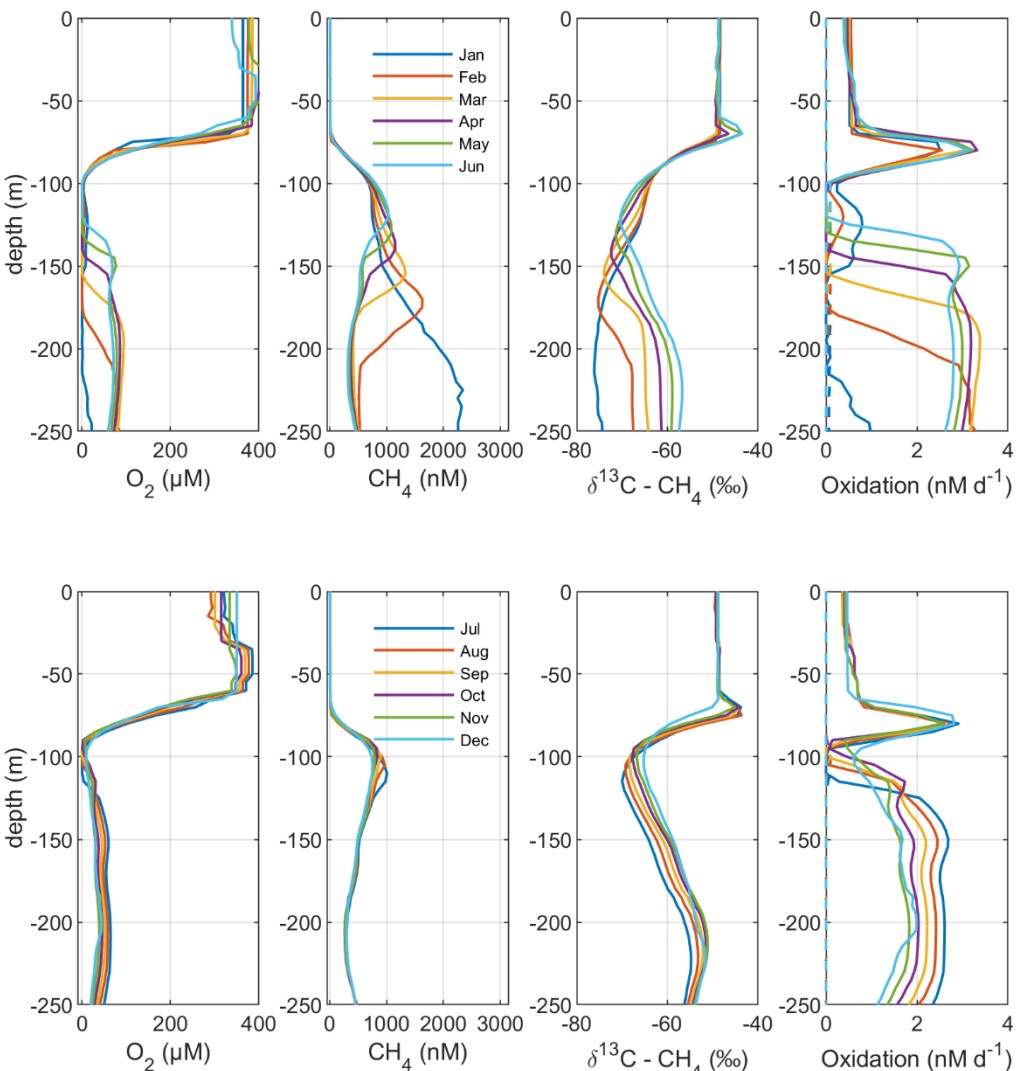

**Figure 5: Model output from the standard model run, showing simulated monthly mean profiles of CH₄ (nM), δ¹³C-CH₄ (‰), and oxidation rates (nM d⁻¹; MOX – full lines, AOM – dashed lines) in the Gotland Sea sub-basin (cf. Fig. S1) year 2015. The upper panels illustrate monthly mean profiles from January to June; the lower panels illustrate monthly mean profiles from July to December.**

Observations from 2012 indicate CH₄ concentrations in a range from ~1000 to 3000 nM in stagnant deep waters of the Baltic Proper (Jakobs et al., 2013; Ketzer et al., 2024), and values below ~150 nM in oxic deep waters in the same area after a major deep-water intrusion in winter 2014–2015 (Schmale et al., 2016; Myllykangas et al., 2017). These values are well reproduced

by the model (Fig. 3) using the settings listed in Table 1, which implies that the simulated sediment $CH_4$ source is likely close to the real source, at least in the deep water where AOM rates are apparently very low (<0.1 nM d$^{-1}$; Jakobs et al., 2014). The simulated benthic $CH_4$ release from the seafloor amounts to 50 µmol $CH_4$ m$^{-2}$ d$^{-1}$ in the standard model run (Table 1), corresponding to ~18 mmol $CH_4$ m$^{-2}$ y$^{-1}$. This is in the lower range of yearly observations at shallow coastal sites among varying habitats in the Baltic Sea (~21-34 mmol $CH_4$ m$^{-2}$ y$^{-1}$; Roth et al., 2023).

Measurements from the central Gotland Sea indicate typical surface water $CH_4$ concentrations in a range 3.5-5 nM, depending on the season (Gülzow et al., 2013), with the highest concentrations observed in winter because of increased gas solubility in cold water. This seasonal cycle is reproduced by the model (Fig. 3). Furthermore, simulated surface water $CH_4$ saturation levels vary between approximately 110% in winter and 150% in summer (Fig. S5), which reproduces observed saturation levels (Gülzow et al., 2013).

Measurements from the central Baltic Sea indicate MOX rates ranging from 0.1 to 4 nM d$^{-1}$ in the redoxcline (Schmale et al., 2012; 2016; Jakobs et al., 2014). In the standard model run, the highest oxidation rates (> 3 nM d$^{-1}$; Fig. 5) occur in the deep water after deep water intrusions leading to oxygenation of stagnant water with high $CH_4$ concentrations. In the redoxcline, the simulated MOX rates are typically in a range 0.5-3 nM d$^{-1}$ (Fig. 4-5), which thus matches observed oxidation rates. Simulated surface water MOX rates are in a range 0.3-0.5 nM d$^{-1}$ (e.g., Fig. 3), whereas observations, on the other hand, indicate rates close to zero (Jakobs et al., 2014).

Observations indicate a pronounced $^{13}$C-$CH_4$ enrichment in the redoxcline. Based on two profiles from 2012, $\delta^{13}$C-$CH_4$ increased from values below -70‰ at the bottom of the redoxcline (~140 m) to -40‰ at the top of the redoxcline (~80 m) in the central Gotland Sea (Jakobs et al., 2014). The $\delta^{13}$C-$CH_4$ peak values at intermediate depths coincide with peak oxidation rates (Jakobs et al., 2014) and result from the preferential oxidation of the lighter isotope. In water above the top of the redoxcline, observations indicate lower oxidation rates and $\delta^{13}$C-$CH_4$ values in a range -60‰ to -40‰ depending on season (Jakobs et al., 2014). In the standard model run, the $\delta^{13}$C-$CH_4$ value typically increases from approximately -70‰ at the upper limit for anoxic water (~130 m) to its peak values between -45‰ and -40‰ at approximately 75 m (Fig. 4). The simulated $\delta^{13}$C-$CH_4$ in the redoxcline thus tends to be less pronounced than what is apparent from the few available observations. Furthermore, a local minimum around 30 m observed by Jakobs et al. (2014) is not reproduced in the model run (see further discussion in Sect. 4).

## 3.2 Preliminary CH$_4$ budget

Here, we present preliminary budget calculations based on the standard model run. It is however important to stress that these estimates are heavily dependent on the prescribed benthic $CH_4$ source. As discussed below (Sect. 4), different combinations of benthic $CH_4$ release and MOX rates could produce similar $CH_4$ concentrations in the water column.

To allow a preliminary assessment of the relative importance of different processes, total $CH_4$ sources (river load, import from adjacent sub-basins, and sediment release) and sinks (outgassing, export to adjacent sub-basins, and pelagic oxidation) were aggregated over the Baltic Proper (sub-basin 7-9, Figure S1), representing the area where the model has been fitted based on

available observations. The CH$_4$ sources were largely dominated by benthic release which amounted to an average 4155 Mmol y$^{-1}$ over the 2001-2020 period (Table 2). This source was mainly balanced by oxidation in the water column (3816 Mmol y$^{-1}$, 92% of the sinks) and to a smaller degree by emission to the atmosphere (348 Mmol y$^{-1}$, 8% of the sinks). The river load (11 Mmol y$^{-1}$) and net exchange (import - export) with adjacent sub-basins (8 Mmol y$^{-1}$) were comparatively small.

Table 2. Total CH$_4$ sources, sinks, and net change (= sources - sinks) (Mmol y$^{-1}$) aggregated over the Baltic Proper (sub-basin 7-9, Figure S1) and averaged over the period 2001-2020.

| CH$_4$ sources, sinks, and net change | CH$_4$ flux (Mmol y$^{-1}$) |
| --- | --- |
| River load | 11 |
| Air-sea exchange | -348 |
| Import - export | 8 |
| Pelagic oxidation | -3816 |
| Sediment release | 4155 |
| Net change | 10 |

Figure 6 illustrates simulated monthly fluxes, net accumulation as well as the total amount of CH$_4$ in the Baltic Proper. The total CH$_4$ stock amounted to almost 1800 Mmol over the ~2010-2014 period, which exceeded the stock before and after that period by a factor 3 (Fig. 6). This comparatively large CH$_4$ stock was the result of a large anoxic deep-water volume and thus low oxidation rates (Fig. 2). There was an average net accumulation of 10 Mmol y$^{-1}$ over the 2001-2020 period (Table 2), but net changes of the total CH$_4$ stock between individual years varied considerably, which largely reflected oxygen dependent changes in CH$_4$ oxidation rates (Fig. 6).

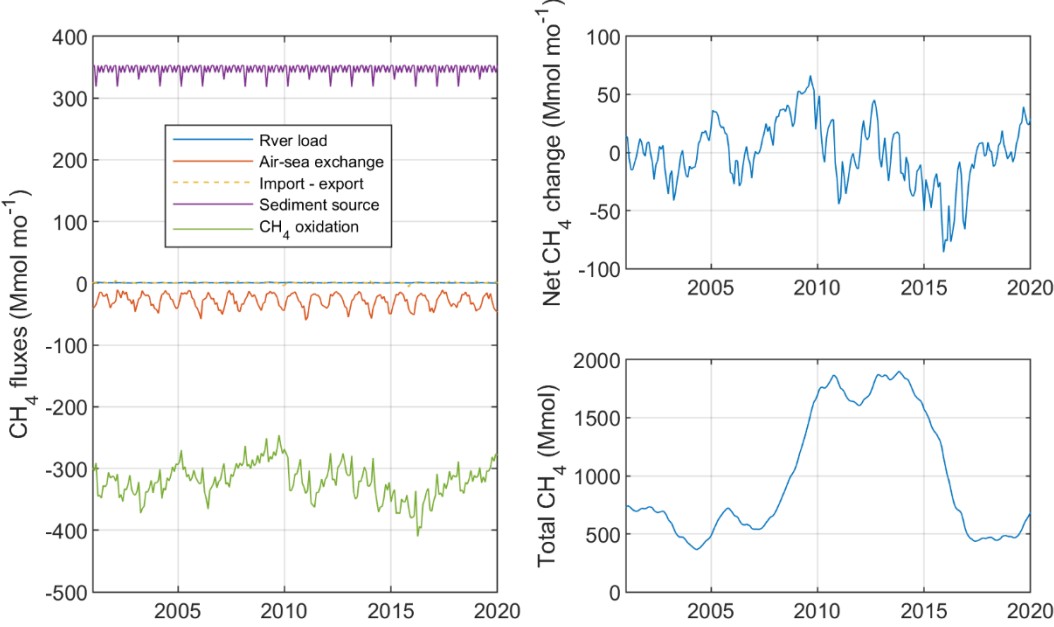

**Figure 6: Simulated monthly mean CH₄ sources and sinks (Mmol mo⁻¹; left), net change (= sources - sinks) (Mmol mo⁻¹; upper right), and total CH₄ stock (Mmol; lower right) aggregated over the Baltic Proper (sub-basin 7-9, Figure S1).**

## 4 Discussion

This study presents a first quantification of key CH₄ fluxes in the Baltic Proper. However, there are uncertainties in our
estimates, in particular regarding the benthic CH₄ source. In the standard model run, benthic release is the dominant CH₄ source
(Table 2). The sediment source is set as constant over time, at all depths, and in all sub-basins. In the real Baltic Sea, however,
large spatial and temporal variations are expected (e.g., Roth et al., 2022). Furthermore, the isotopic composition of the
sediment source is set either to -80‰ or -60‰ depending on oxygen conditions in the overlying water. This assumption is a
simplified representation. The main uncertainty in our present large-scale estimates is that spatial and temporal variations of
the sediment source are not well known.

The simulated CH₄ concentrations in anoxic deep waters agree with available observations. The fitted rate of CH₄ release from
sediments is for that reason deemed as feasible in anoxic deep waters, since CH₄ concentrations are only marginally influenced
by oxidation during anoxic conditions (low AOM rates). It is, however, likely that the fitted CH₄ release is mainly
representative for present-day conditions (e.g., organic carbon deposition rates, oxygen concentrations, temperatures, etc.).
Both climate change and nutrient load change are going to affect e.g., oxygen concentrations in the future which means that
the benthic CH₄ source is likely to change as well. In order to address this, it is necessary to improve the knowledge of CH₄

release rates depending on local conditions. One major uncertainty here is what is the contribution from more recent organic carbon deposition, and what is the contribution from "old" carbon deeper down in the sediments, i.e., if nutrient loads and organic carbon deposition decreases, and oxygen conditions improve, would this have a major impact on the $CH_4$ release from sediments, or is the release more heavily dependent on older carbon deposits? This is one of the major remaining open questions regarding $CH_4$ cycling in the Baltic Sea, but cannot be addressed by the model at this point.

While the fitted flux gives a good idea of the present-day $CH_4$ source in deeper areas, it is more challenging to constrain the sediment source in shallower oxic waters, where the source can be largely compensated by MOX in the water column. Coastal systems are also more dynamic and show a larger variety compared to deep anoxic areas. A large $CH_4$ source compensated by high MOX rates could for example yield similar $CH_4$ concentrations as a smaller source combined with lower MOX rates. These two different cases (i.e., large source, high oxidation vs. small source, low oxidation) would produce quite different isotopic patterns that could be used to calibrate the model. However, a complication here is that we generally do not know the isotopic composition of $CH_4$ released from the sediments, with the exception of observational data from a few locations. Justification of the fitted rates used in the model would require more observational data to fill the knowledge gaps.

Studies from wetlands (Segers, 1998), lakes (Martinez-Cruz et al., 2015; Tan et al., 2015), and oceanic sites (Kessler et al., 2011; Crespo-Medina et al., 2014; Pack et al., 2015; Rogener et al., 2018; Chan et al., 2019a) show that MOX rates can vary by several orders of magnitude. For example, observed deep water MOX in the Gulf of Mexico increased from a background rate of around 60 pM d$^{-1}$ to a peak rate of 5900 nM d$^{-1}$ after the Deepwater Horizon oil spill (Rogener et al., 2018). The observed MOX rates from the central Baltic Sea (approximately 0.1-4 nM d$^{-1}$; Schmale et al., 2012; 2016; Jakobs et al., 2014) are in the same range as MOX rate observations from the eastern tropical North Pacific Ocean (Pack et al., 2015), but typically lower than MOX rates observed in lakes (Martinez-Cruz et al., 2015; Tan et al., 2015).

In this study, the parameter values used in the computation of MOX rates (Eq. 16) were fitted so that the resulting profiles of oxidation rates and isotopic composition – as well as $CH_4$ concentrations – reasonably well reproduce observed profiles from the central Baltic Sea. Results for $CH_4$ concentrations, MOX rates, and isotopic composition are sensitive to $O_2$ profiles, which also means that the fitted values depend on how well the model reproduces $O_2$ concentrations.

The rate constant for MOX depends on the activity and abundance of methanotrophs, in theory allowing for reduced MOX in spite of favorable conditions in terms of $CH_4$ and $O_2$ concentrations when methanotrophs are not active. The model does not include methanotrophic activity explicitly, and the rate constant for MOX is constant. Perhaps, lower abundance and activity of methanotrophs could be an explanation for the lower rate constant in the present results compared to the results from lakes cited above.

The present study does not include the potential contributions from aerobic $CH_4$ production. There are, however, several potential pathways for $CH_4$ production in shallow oxic waters, including e.g., direct $CH_4$ production by phytoplankton (Lenhart et al., 2016) and cyanobacteria (Bižić et al., 2020), $CH_4$ production as a biproduct of microbial degradation processes (Karl et al., 2008; Damm et al., 2010), and $CH_4$ formation in anoxic microniches within degrading detritus (Karl and Tilbrook, 1994; Holmes et al., 2000). In the Baltic Sea, local $CH_4$ maxima coinciding with $\delta^{13}C$-$CH_4$ minima have been observed in oxic waters

just below the summer thermocline (Jakobs et al., 2014; Schmale et al., 2018). These signals can be coupled to zooplankton grazing activities, both directly through $CH_4$ production during digestion, and indirectly via release of methanogenic substrates that can subsequently be degraded to methane by microbes (Schmale et al., 2018; Stawiarski et al., 2019). However, the main pathways as well as magnitude of aerobic $CH_4$ production in the Baltic Sea remain to be resolved in detail. Parameterizations of these processes can then potentially be included in models such as BALTSEM that explicitly include both phytoplankton and zooplankton groups as model state variables.

## 4.1 Sensitivity experiments

A series of sensitivity experiments was performed on different parameters used in the modeling of $CH_4$ and its stable isotopes (Table 1). The adjusted parameter values are listed in Table 3. Modeled profiles are then drawn for both winter conditions (February) and summer conditions (August) of 2015 (Fig. S6-S11), which gives an indication of season dependent contrasting conditions in surface waters above the halocline. Methane cycling in the model is largely dominated by benthic release, oxidation in the water column, and outgassing (Table 2; Fig. 6). For that reason, sensitivity experiments on riverine and North Sea $CH_4$ concentrations were not included.

Table 3. Adjusted parameter values and change (%) compared to the standard model run in the various sensitivity experiments.

| Model run | Adjusted parameter | Notation | Value | Unit |
|---|---|---|---|---|
| test 1 | Potential maximum oxidation rate (MOX) | $v_{WCH4\_O2}$ | 4 (-50%) | nM d$^{-1}$ |
| test 2 | Potential maximum oxidation rate (MOX) | $v_{WCH4\_O2}$ | 12 (+50%) | nM d$^{-1}$ |
| test 3 | Half saturation value, $CH_4$ oxidation | $h_{CH4}$ | 30 (-50%) | nM |
| test 4 | Half saturation value, $CH_4$ oxidation | $h_{CH4}$ | 90 (+50%) | nM |
| test 5 | Half saturation value, $CH_4$ oxidation | $h_{O2}$ | 50 (-50%) | µM |
| test 6 | Half saturation value, $CH_4$ oxidation | $h_{O2}$ | 150 (+50%) | µM |
| test 7 | Fractionation, $CH_4$ oxidation | $\alpha_{oxi}$ | 0.984 (-4‰) | - |
| test 8 | Fractionation, $CH_4$ oxidation | $\alpha_{oxi}$ | 0.992 (+4‰) | - |
| test 9 | Sediment source, $\delta^{13}C$-$CH_4$, oxic water | $\delta^{13}C$-$CH_{4sed}$ | -70 (-10‰) | ‰ |
| test 10 | Sediment source, $\delta^{13}C$-$CH_4$, oxic water | $\delta^{13}C$-$CH_{4sed}$ | -50 (+10‰) | ‰ |
| test 11 | Sediment source, $CH_4$ flux, oxic water | $r_{SED}$ | 25 (-50%) | µmol m$^{-2}$ d$^{-1}$ |
| test 12 | Sediment source, $CH_4$ flux, oxic water | $r_{SED}$ | 75 (+50%) | µmol m$^{-2}$ d$^{-1}$ |

### 4.1.1 MOX and AOM: rates, half-saturation constants, and fractionation

Adjusting the potential maximum rate of MOX ($v_{WCH4\_O2}$) by $\pm$ 50% (tests 1-2) has a large influence on $CH_4$ concentrations (Fig. S6), where decreased $v_{WCH4\_O2}$ (test 1) leads to substantially higher $CH_4$ concentrations, and increased $v_{WCH4\_O2}$ (test 2) to lower $CH_4$ compared to the standard model run. Since the MOX rate in addition to $v_{WCH4\_O2}$ depends on $CH_4$ concentration, the changed $CH_4$ concentration in itself will further modify the shape of the MOX profile ($CH_4$ oxidation also consumes $O_2$, but the influence on $O_2$ concentration is small compared to the influence on $CH_4$ concentration, since $O_2$ and $CH_4$ typically differ by orders of magnitude). The modified shapes of the MOX profiles also influence the $\delta^{13}C$-$CH_4$ profiles, with changed depths of the intermediate deep-water peak as well as changed peak values. Adjusting the potential maximum rate of AOM ($v_{WCH4\_SO4}$) has a comparatively minor influence on both $CH_4$ concentration and isotopic composition because of the low anaerobic oxidation rates (not shown).

Adjusting the half saturation values for $CH_4$ oxidation ($h_{CH4}$ and $h_{O2}$) by $\pm$ 50% (tests 3-6) influences the MOX rates and thus both the $CH_4$ concentration and the isotopic composition (Fig. S7-S8). These parameters alter the dynamics within a relatively small range close to their respective values. Thus, the MOX rate is most sensitive to changes of $h_{CH4}$ where the $CH_4$ concentration is close to 60 nM, and similarly, most sensitive to changes of $h_{O2}$ where the $O_2$ concentration is close to 100 $\mu$M (Table 1). At high concentrations compared to the values of $h_{CH4}$ and $h_{O2}$, we do not expect a large impact by adjusting these constants. On the other hand: at low concentrations compared to the constants, the sensitivity to changed values of $h_{CH4}$ and $h_{O2}$ is expected to be similar to changing the potential maximum rate constant ($v_{WCH4\_O2}$).

In these particular experiments, $CH_4$ dynamics are more sensitive to changes in $h_{O2}$ than $h_{CH4}$, and the reason for this is the relatively large water volume where the $O_2$ concentration is close to $h_{O2}$, while the $CH_4$ concentration on the other hand is only close to $h_{CH4}$ in a comparatively narrow band at intermediate depths. The modified $CH_4$ dynamics are, however, transferred to other depths by turbulent diffusion and vertical internal circulation ('old' water mixing into the intruding new deep water), which means that altered $CH_4$ concentrations, $\delta$ values, and MOX rates are (more or less) apparent throughout the entire water column.

Adjusting the fractionation during $CH_4$ oxidation by $\pm$ 4‰ (tests 7-8) has no influence on $CH_4$ oxidation rates and concentrations, but a relatively strong (and predictable) impact on $\delta^{13}C$-$CH_4$ values throughout the entire water column (Fig. S9).

### 4.1.2 Sediment source: CH₄ release and isotopic composition

As indicated in Sect. 4, it is expected that the isotopic composition of the sediment source differs between different locations depending on the degree of oxidation in the pore water. The rate of $CH_4$ release is also expected to depend on the balance between benthic $CH_4$ production and oxidation, respectively. Adjusting the $\delta^{13}C$-$CH_4$ value of the sediment source by $\pm$ 10‰ during oxic conditions (tests 9-10) has no influence on $CH_4$ oxidation rates and concentrations, but a strong (and predictable) impact on $\delta^{13}C$-$CH_4$ profiles (Fig. S10).

In experiments where the rate of CH$_4$ release from the sediments source was adjusted by $\pm$ 50‰ during oxic conditions (tests 11-12), strong impacts are apparent for both the CH$_4$ concentration and isotopic composition throughout the water column. Deep water MOX rates are however less sensitive since the rates in these cases depend more on O$_2$ concentration – which is very similar between the two experiments (not shown) – than CH$_4$ concentrations (Fig. S11).

## 4.2 Caveats and outlook

As previously discussed, the main uncertainty in the model simulations lies in our limited understanding of CH$_4$ release from different sediment areas, as well as the isotopic composition of CH$_4$ released into the water column. Both the flux and the isotopic composition depend on the balance between production and oxidation rates in the sediment. A high production could be compensated by high oxidation and thus result in a relatively small CH$_4$ release to the water column in spite of a large production. This would then be evident by a $^{13}$C-CH$_4$ enrichment, i.e., comparatively heavy CH$_4$. Alternatively, a relatively small CH$_4$ production could still result in a substantial release to the water column in a case where the oxidation rate is low, which would then also be evident by CH$_4$ depleted in $^{13}$C-CH$_4$, i.e., comparatively light CH$_4$.

Improved knowledge of properties of CH$_4$ released from sediment to water column in different areas of the Baltic Sea (e.g., the Kattegat and the major gulfs – the Gulf of Bothnia, Gulf of Riga, and Gulf of Finland) would help to improve model parameterizations and thus reduce the main uncertainties of model simulations. This was, however, beyond the scope of the present study because of the missing knowledge concerning both temporal and spatial patterns of the CH$_4$ source. A logical progression at this stage would involve detailed observations combined with modeling studies focused on processes in the sediments, i.e., production and oxidation rates, depending on carbon accumulation rate, oxygen conditions, and the presence of methanotrophs.

A crucial missing link in this study is the formation, transport, and fate of CH$_4$ bubbles. Estimates by Weber et al. (2019) indicate that ebullitive fluxes contribute a major fraction of CH$_4$ released to the atmosphere from shallow coastal areas. Ebullition events have been observed in the Baltic Sea, both at coastal sites (e.g., Humborg et al., 2019; Lohrberg et al., 2020; Lehoux et al., 2021; Hermans et al., submitted) and deep water accumulation bottoms (C. Stranne, unpublished data). Ebullition has been included in lake models (e.g., Greene et al., 2014; Stepanenko et al., 2016; Bayer et al., 2019); however, we do not have experimental data to calibrate and validate the large-scale influence of ebullition in the Baltic Sea. The fitted benthic CH$_4$ source represents a "bulk" CH$_4$ release, including in theory both the influences of diffusive flux and bubble dissolution on CH$_4$ concentrations in the water column. However, CH$_4$ ebullition might bypass methanotrophy and consequently contribute to higher CH$_4$ emissions, in particular in shallow-water areas (e.g., Broman et al., 2020). This indicates that the simulated CH$_4$ outgassing is likely underestimating the real outgassing from the Baltic Sea. Observations of ebullitive fluxes in combination with development of model parameterizations represent important steps to better describe and quantify CH$_4$ emissions from the Baltic Sea. When it comes to local production of gas bubbles and the transformation and fate of methane in the bubbles, the horizontally averaged approach used in the present study is most likely insufficient, which could be addressed either by 3D modelling or by adding smaller sub-domains to the present model.

Roth et al. (2023) observed significant $CH_4$ production and release from vegetated oxic shallow-water areas. BALTSEM-$CH_4$ v1.0 does not differentiate between vegetated and unvegetated areas, which means that this $CH_4$ source – and its contribution to outgassing – could not be addressed here, which consequently represents another gap in our current understanding. Both species distribution models and process-based models for vegetation exist (e.g., Lappalainen et al., 2019; Graiff et al., 2020), but to our knowledge do not include $CH_4$ dynamics. Hence, the inclusion of $CH_4$ in vegetation models could potentially serve as an objective for future scientific projects.

The process parameterizations used in this study to describe large-scale $CH_4$ cycling in the Baltic Sea can also be applied in various other domains. As part of our future plans, we aim to investigate $CH_4$ dynamics in a smaller area where more observations are available and where the $CH_4$ concentration and isotopic composition, as well as properties of end-members (river load, benthic release, and lateral boundary conditions), are better understood. This would further help to constrain process rates in the model.

The calculated average total $CH_4$ emission of 348 Mmol $y^{-1}$ from the Baltic Proper corresponds to approximately 1.5 mmol $CH_4$ $m^{-2}$ $y^{-1}$, and constitutes only about 8% of the fitted sediment source (~18 mmol $CH_4$ $m^{-2}$ $y^{-1}$). The model includes both shallow- and deep water sediment areas, but the fitted sediment source is in the lower range of rates reported for a shallow-water coastal area (~21-34 mmol $CH_4$ $m^{-2}$ $y^{-1}$; Roth et al., 2023), indicating that the model might not well represent coastal $CH_4$ hotspots. One major knowledge gap at this point is the relative importance of shallow coastal areas compared to the open Baltic Sea in terms of $CH_4$ outgassing. This is an important scientific question that needs to be addressed in future studies.

**Code and data availability**

All model output data for the standard model run, as well as the version of the model source code used in this study are archived on Zenodo at https://doi.org/10.5281/zenodo.10037197 (last access: 24 October 2023). In addition to model output and source code, the archive includes initial profiles, boundary conditions, meteorological forcing, runoff, as well as river loads, point sources, and atmospheric depositions of dissolved and particulate constituents.

**Author contributions**

E.G. wrote the first draft of the manuscript with contributions from all co-authors. E.G. developed the $CH_4$ model components, and designed and performed the model experiments and analyses. B.G. is the main developer of the BALTSEM model.

**Competing interests**

The contact author has declared that none of the authors has any competing interests.

## Acknowledgements

This research is part of the University of Helsinki and Stockholm University collaborative research initiative (CoastClim, www.coastclim.org; The Baltic Bridge initiative). Financial support was provided by the Swedish Research Council (VR) grant 2021-04641. The Baltic Nest Institute is supported by The Swedish Agency for Marine and Water Management through their grant 1:11 – Measures for marine and water environment.

## Appendix A: State variables and biogeochemical transformation processes

Table A1. Pelagic and sediment state variables in BALTSEM-CH$_4$ v1.0.

| State variable | Description | Unit |
|---|---|---|
| *Pelagic* | | |
| SAL | Salinity | - |
| T | Temperature | °C |
| OXY | Dissolved oxygen | g O$_2$ m$^{-3}$ |
| NH | Ammonium | mg N m$^{-3}$ |
| NO | Nitrate + nitrite | mg N m$^{-3}$ |
| PO | Phosphate | mg P m$^{-3}$ |
| SiO | Dissolved silica | mg Si m$^{-3}$ |
| DETN | Detrital N | mg N m$^{-3}$ |
| DETP | Detrital P | mg P m$^{-3}$ |
| DETSi | Detrital Si | mg Si m$^{-3}$ |
| DETCm | Detrital C (autochthonous) | mg C m$^{-3}$ |
| DETCt | Detrital C (allochthonous) | mg C m$^{-3}$ |
| PHY1 | Phytoplankton group 1, N$_2$ fixers | mg N m$^{-3}$ |
| PHY2 | Phytoplankton group 2, diatoms | mg N m$^{-3}$ |
| PHY3 | Phytoplankton group 3, other phytoplankton | mg N m$^{-3}$ |
| ZOO | Heterotrophs/zooplankton | mg N m$^{-3}$ |
| DONL | Labile dissolved organic N | mg N m$^{-3}$ |
| DONR | Refractory dissolved organic N | mg N m$^{-3}$ |
| DOPL | Labile dissolved organic P | mg P m$^{-3}$ |
| DOPR | Refractory dissolved organic P | mg P m$^{-3}$ |
| DOCLt | Labile dissolved organic C (allochthonous) | mg C m$^{-3}$ |
| DOCRt | Refractory dissolved organic C (allochthonous) | mg C m$^{-3}$ |

| DOCLm | Labile dissolved organic C (autochthonous) | mg C m$^{-3}$ |
| DOCRm | Refractory dissolved organic C (autochthonous) | mg C m$^{-3}$ |
| DIC | Dissolved inorganic carbon | mmol m$^{-3}$ |
| ALK | Total alkalinity | mmol m$^{-3}$ |
| HS | Hydrogen sulfide | mg S m$^{-3}$ |
| 12CH4 | $^{12}$C methane | µmol m$^{-3}$ |
| 13CH4 | $^{13}$C methane | µmol m$^{-3}$ |

*Sediment*

| SEDN | Sedimentary organic N | mg N m$^{-2}$ |
| SEDP | Sedimentary organic P | mg P m$^{-2}$ |
| SEDSi | Sedimentary organic Si | mg Si m$^{-2}$ |
| SEDCm | Sedimentary organic C (autochthonous) | mg C m$^{-2}$ |
| SEDCt | Sedimentary organic C (allochthonous) | mg C m$^{-2}$ |

**Appendix B: Model forcing**

Model forcing consists of actual weather data and observed nutrient loads as well as calibrated carbon and total alkalinity loads (Gustafsson and Gustafsson, 2020) covering the period 1970-2020. River runoff, land loads, and atmospheric depositions were
based on Pollution Load Compilation data (PLC; HELCOM, 2021) as well as other sources (Gustafsson et al., 2012). Atmospheric forcing was constructed from data provided by the Swedish Meteorological and Hydrological Institute (SMHI): RCA-ERA40 (1970-2006), Hirlam-Mesan (2007-2015), and Arome-Mesan (2016-2020). The Kattegat water level and also boundary conditions in the Skagerrak were based on data provided by the SMHI (Gustafsson et al., 2012).

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
