# Peer review of "Methane dynamics in the Baltic Sea: investigating concentration, flux and isotopic composition patterns using the coupled physicalbiogeochemical model BALTSEM-CH4 v1.0"

_Geoscientific Model Development, 2023_

## Author Response (AR2)

Author's point-by-point response

The revised manuscript includes several new paragraphs of text according to suggestions by the two referees, as well as several new references.

All relevant changes to the manuscript are described in detail in the responses to reviewers, as indicated in the point-by-point response below:

**Response to reviewer 1:**

Review: "Methane dynamics in the Baltic Sea: investigating concentration, flux, and isotopic composition patterns using the coupled physical, biogeochemical model BALTSEM-CH4 v1.0" by E. Gustafsson, B. G. Gustafsson, M. Hermans, C. Humborg, and C. Stranne, under review for Geoscientific Model Development

As a very short summary, reactions and tracers were added to an ecosystem model that includes the horizontally integrated depth profiles of model tracers in 13 sub-basins of the Baltic Sea to simulate methane concentrations and isotopes and resolve the associated process rates. The baseline simulation shows high CH4 accumulation in the deeper anoxic parts of the Baltic Sea. Redox zonation is strongly affected over time by inflows from the North Sea and stagnation periods. CH4 concentrations above the halocline are generally much lower due to the presence of oxygen. There, seasonal thermal stratification plays an important role due to its effect on O2 availability and CH4 and O2 solubility. This also strongly affects the d13C-CH4 profiles. Qualitative agreement exists between simulated 1-D profiles, representing larger 3-D volumes of the sub-basins, and measured profiles (e.g., Jakobs et al., 2014). The authors also present a preliminary methane budget, which is, however, highly uncertain.

The introduction does a good job of describing the main research questions for studying methane dynamics in the Baltic Sea. It introduces a problem that the 1-D model cannot resolve, viz., lateral CH4 concentrations from the coastal to open waters. Similarly, there is a significant discussion in the manuscript about benthic methane release from shallower parts that cannot be well resolved. The authors could have chosen to add CH4 isotope tracers and reactions to existing 3-D ecosystem models instead of BALTSEM. This would have allowed simulating lateral gradients and also direct comparison of vertical simulated profiles from deeper parts to measured profiles at specific locations. The authors mention that BALTSEM was chosen since it has been calibrated in the past. However, other 3-D ecosystem models have also been calibrated for the Baltic Sea. Given the link between primary production and sedimentary methane production and the importance of shallow point sources of methane, it may seem that a budget including methane emission may only be confidently constrained with a 3-D model. The authors could elaborate on this choice between 1-D and 3-D models.

*Response: We have not included a general discussion regarding the pros and cons of 1D versus 3D modelling approaches in the text, since this would derail a bit from the scope of the manuscript. For the purposes of the present study – i.e., parameterizations of processes in the cycling of methane, and a first preliminary budget on a Baltic Sea scale – the use of a horizontally averaged model gives a good idea about the overall methane fluxes. That said, the main issue is not that BALTSEM does not resolve horizontal gradients within the sub-basins, the issue is rather our very limited understanding of how e.g., the benthic methane source and its isotopic composition varies in time and space depending on local conditions*

*(e.g., organic carbon deposition rate, oxygen concentration, temperature, etc.). A 3D model would produce horizontal gradients of methane concentrations, but model validation would still be limited by knowledge gaps for some of the key processes.*

*It may however be noted that what appears to be of particular importance here is to improve the knowledge of contributions from ebullitive fluxes, since methane bubbles – as opposed to the diffusive flux – can bypass stratification and thus produce enhanced sea-air fluxes. Considering the very local scales of ebullition events and bubble trains, this is a case where the 1D approach in the present study is likely to be insufficient.*

*The following text has been added in Section 4.2, line 682-684:*

*"When it comes to local production of gas bubbles and the transformation and fate of methane in the bubbles, the horizontally averaged approach used in the present study is most likely insufficient, which could be addressed either by 3D modelling or by adding smaller sub-domains to the present model."*

*One major next step to improve the understanding and close some of the knowledge gaps, is to focus future efforts on the methane dynamics at coastal sites in general, and on ebullitive fluxes in particular – both in terms of measurements and modelling. This is discussed in Section 4.2, line 679-682:*

*"This indicates that the simulated CH4 outgassing is likely underestimating the real outgassing from the Baltic Sea. Observations of ebullitive fluxes in combination with development of model parameterizations represent important steps to better describe and quantify CH4 emissions from the Baltic Sea."*

*It is still an open question how important high concentrations and high sea-air fluxes from small shallow areas are compared to low concentrations and fluxes from large open areas.*

In the manuscript, there is great uncertainty regarding the sources of methane (acknowledged in the discussion). The paper assumes a constant flux from the sediment to the water column, independent of time or water depth. There are a couple of issues, including some that are not discussed.

First of all, the text does not elaborate on sources of methane in the water column, which could be particularly important for methane emissions. Studies have shown that methane can be produced in oxic water. Weber et al. (2019) argue that these pathways are needed to explain the general oversaturation of methane in ocean surface waters, and they mention a strong correlation between methane production and net primary production. Could methane production in the water column play an important role in the Baltic Sea? Could the degradation of methylphosphonate form an alternative explanation for higher CH4 emissions (instead of lower CH4 solubility and increased benthic methanogenesis) in exceptionally warm summers, wherein PO4 is more likely a strong limiting factor for primary production? Coccolithophores and zooplankton can also release methane in the water column. Anoxic microzones in sinking particles could harbor methanogenesis both in surface waters and deeper waters. The paper mentions that the model does not reproduce a measured local

minimum d13C-CH4 at 30 m depth in the water column (line 431). Local production of methane could potentially explain this.

*Response: The processes behind aerobic CH4 production are still not resolved in the Baltic Sea, although recent evidence suggests links to zooplankton grazing (as briefly mentioned in Section 4). The text in Section 4 has now been replaced by the following expanded discussion on potential unresolved CH4 sources (Section 4, lines 599-609):*

*"The present study does not include the potential contributions from aerobic $CH_4$ production. There are, however, several potential pathways for $CH_4$ production in shallow oxic waters, including e.g., direct $CH_4$ production by phytoplankton (Lenhart et al., 2016) and cyanobacteria (Bižić et al., 2020), $CH_4$ production as a biproduct of microbial degradation processes (Karl et al., 2008; Damm et al., 2010), and $CH_4$ formation in anoxic microniches within degrading detritus (Karl and Tilbrook, 1994; Holmes et al., 2000). In the Baltic Sea, local $CH_4$ maxima coinciding with $\delta^{13}C$-$CH_4$ minima have been observed in oxic waters just below the summer thermocline (Jakobs et al., 2014; Schmale et al., 2018). These signals can be coupled to zooplankton grazing activities, both directly through $CH_4$ production during digestion, and indirectly via release of methanogenic substrates that can subsequently be degraded to methane by microbes (Schmale et al., 2018; Stawiarski et al., 2019). However, the main pathways as well as magnitude of aerobic $CH_4$ production in the Baltic Sea remain to be resolved in detail. Parameterizations of these processes can then potentially be included in models such as BALTSEM that explicitly include both phytoplankton and zooplankton groups as model state variables."*

Secondly, fluxes from sediment to the water are likely not constant. Clearly, the authors are well aware of seasonal variations in CH4 effluxes (lines 458-464) and also that it strongly depends on the oxygen levels in bottom waters (Reed et al., 2011). This makes the simulated temporal patterns less reliable. The discussion and sensitivity analyses sufficiently address this issue and, indeed, show a high sensitivity toward the parameter used for benthic CH4 release.

*Response: Yes, as discussed in the manuscript the main uncertainty in this study is the CH4 release from sediment to water: this flux is known to be highly variable both in space and time (e.g., Roth et al., 2022) as a function of organic carbon deposition rate, oxygen and sulfate concentrations, etc. Observed fluxes are very sparse which means that it is a major challenge at this point to parameterize the CH4 effluxes from different sediment areas of the Baltic Sea. The calibrated constant flux gives a very good idea about the long-term mean CH4 release from deep water sediments in stagnant waters where CH4 in the water seems to be only marginally influenced by other processes (e.g., AOM very low in anoxic waters as demonstrated by measurements by e.g., Jakobs et al., 2013).*

*The following text was added to Section 4, line 565-575:*

- *"It is, however, likely that the calibrated $CH_4$ release is mainly representative for present-day conditions (e.g., organic carbon deposition rates, oxygen concentrations, temperature, etc.). Both climate change and nutrient load change are going to affect e.g., oxygen concentrations in the future which means that the benthic $CH_4$ source is likely to change as well. In order to address this, it is necessary to improve the knowledge of $CH_4$ release rates depending on local conditions. One major uncertainty here is what is the contribution from more recent organic carbon deposition, and*

*what is the contribution from "old" carbon deeper down in the sediments, i.e., if nutrient loads and organic carbon deposition decreases, and oxygen conditions improve, would this have a major impact on the CH$_4$ release from sediments, or is the release more heavily dependent on older carbon deposits? This is one of the major remaining open questions regarding CH$_4$ cycling in the Baltic Sea, but this cannot be addressed by the model at this point."*

- *"While the calibrated flux gives a good idea of the present-day CH$_4$ source in deeper areas, it is more challenging to constrain the sediment source in shallower oxic waters, where the source can be largely compensated by MOX in the water column."*

*Furthermore, the uncertainty has previously been addressed in the Section 4 (line 556-562):*

- *"However, there are uncertainties in our estimates, in particular regarding the benthic CH4 source. In the standard model run, benthic release is the dominant CH4 source (Table 2). The sediment source is set as constant over time, at all depths, and in all sub-basins. In the real Baltic Sea, however, large spatial and temporal variations are expected (e.g., Roth et al., 2022). Furthermore, the isotopic composition of the sediment source is set either to -80‰ or -60‰ depending on oxygen conditions in the overlying water. This assumption is a simplified representation. The main uncertainty in our present large-scale estimates is that spatial and temporal variations of the sediment source are not well known."*

*Section 4.2 (line 657-670):*

- *"As previously discussed, the main uncertainty in the model simulations lies in our limited understanding of CH4 release from different sediment areas, as well as the isotopic composition of CH4 released into the water column. Both the flux and the isotopic composition depend on the balance between production and oxidation rates in the sediment. A high production could be compensated by high oxidation and thus result in a relatively small CH4 release to the water column in spite of a large production. This would then be evident by a 13C-CH4 enrichment, i.e., comparatively heavy CH4. Alternatively, a relatively small CH4 production could still result in a substantial release to the water column in a case where the oxidation rate is low, which would then also be evident by CH4 depleted in 13C-CH4, i.e., comparatively light CH4. "*
- *"Improved knowledge of properties of CH4 released from sediment to water column in different areas of the Baltic Sea (e.g., the Kattegat and the major gulfs – the Gulf of Bothnia, Gulf of Riga, and Gulf of Finland) would help to improve model parameterizations and thus reduce the main uncertainties of model simulations. This was, however, beyond the scope of the present study because of the missing knowledge concerning both temporal and spatial patterns of the CH4 source. A logical progression at this stage would involve detailed observations combined with modeling studies focused on processes in the sediments, i.e., production and oxidation rates, depending on carbon accumulation rate, oxygen conditions, and the presence of methanotrophs."*

Thirdly, the introduction mentions gas ebullition versus diffusive sources. This part should be expanded. Other workers have shown that hotspots, such as cold seeps, vents, and mud

volcanoes, are significant for methane emissions on the scale of the global ocean (e.g., Hornafius et al., 1999; Weber et al., 2019). There are plenty of studies about cold seeps in the Baltic Sea. To what extent do the authors expect these to dominate emissions to the atmosphere? Methane from these point sources may be laterally transported and affect the CH4 concentrations in surface waters in large parts of the Baltic Sea. There are also other hotspots, such as inundated peat lands. It would be beneficial for readers to know more about the prevalence of cold seeps and methane-rich sediments in both shallow and deeper parts of the Baltic Sea to gain a sense of their importance in the overall budget. It could be very well that methane emissions to the atmosphere are greatly underestimated by the model, as benthic methane release is primarily fitted to CH4 concentrations in the deeper basins.

*Response: Already discussed in Section 4.2, but the text has now been expanded and reads as follows (Section 4.2, line 671-682): "A crucial missing link in this study is the formation, transport, and fate of $CH_4$ bubbles. Estimates by Weber et al. (2019) indicate that ebullitive fluxes contribute a major fraction of $CH_4$ released to the atmosphere from shallow coastal areas. Ebullition events have been observed in the Baltic Sea, both at coastal sites (e.g., Humborg et al., 2019; Lohrberg et al., 2020; Lehoux et al., 2021; Hermans et al., submitted) and deep water accumulation bottoms (C. Stranne, unpublished data). Ebullition has been included in lake models (e.g., Greene et al., 2014; Stepanenko et al., 2016; Bayer et al., 2019); however, we do not have experimental data to calibrate and validate the large-scale influence of ebullition in the Baltic Sea. The calibrated benthic $CH_4$ source represents a "bulk" $CH_4$ release, including in theory both the influences of diffusive flux and bubble dissolution on $CH_4$ concentrations in the water column. However, $CH_4$ ebullition might bypass methanotrophy and consequently contribute to higher $CH_4$ emissions, in particular in shallow-water areas (e.g., Broman et al., 2020). This indicates that the simulated $CH_4$ outgassing is likely underestimating the real outgassing from the Baltic Sea. Observations of ebullitive fluxes in combination with development of model parameterizations represent important steps to better describe and quantify $CH_4$ emissions from the Baltic Sea."*

More than 90% of the citations appear to be works from scientists who have studied the Baltic Sea. Literature from other parts of the world is largely ignored. Occasionally, the wrong articles are cited. For instance, Broman et al. (2020) did not discover methanotrophy (lines 48-50). Not the latest articles, but the articles that made the initial discoveries should be cited. In Table 1, it would be interesting to compare the values of rate constants to the literature. In the discussion, the reaction kinetics are discussed and compared to lake studies. However, there is also a vast body of literature about methane oxidation in ocean waters (e.g., Chan et al., 2019b, 2019b and Pack et al. 2015 show up in a first search attempt).

*Response:*

- *The sentence has been simplified and the Broman reference removed (line 46-48): "Methane formation in sediments can be substantial, but aerobic and anaerobic oxidation processes can efficiently remove $CH_4$ both in the pore water and water column"*
- *The text on lines 89-91 has been updated and now reads: "Methane cycling has previously been investigated in both lake (e.g., Lopes et al., 2011; Greene et al., 2014; Tan et al., 2015; Stepanenko et al., 2016; Bayer et al., 2019) and ocean (e.g., Nihous and Masutani, 2006; Wåhlström and Meier, 2014; Malakhova and Golubeva, 2022) modeling studies."*

- *The reference list has now been expanded to include studies from wetlands, lakes, and ocean areas. The following text has been added in Section 4, line 583-589: "Studies from wetlands (Segers, 1998), lakes (Martinez-Cruz et al., 2015; Tan et al., 2015), and oceanic sites (Kessler et al., 2011; Crespo-Medina et al., 2014; Pack et al., 2015; Rogener et al., 2018; Chan et al., 2019a) show that MOX rates can vary by several orders of magnitude. For example, observed deep water MOX in the Gulf of Mexico increased from a background rate of around 60 pM d-1 to a peak rate of 5900 nM d-1 after the Deepwater Horizon oil spill (Rogener et al., 2018). The observed MOX rates from the central Baltic Sea (approximately 0.1-4 nM d-1; Schmale et al., 2012; 2016; Jakobs et al., 2014) are in the same range as MOX rate observations from the eastern tropical North Pacific Ocean (Pack et al., 2015), but typically lower than MOX rates observed in lakes (Martinez-Cruz et al., 2015; Tan et al., 2015)."*
- *The following text has been added in Section 2.3.6, line 440-441: "Observations indicate a wide range of fractionation during $CH_4$ oxidation (e.g., $\varepsilon \sim$ 4-30‰, Whiticar (1999); $\varepsilon \sim$ 16-54‰, Chan et al. (2019b))".*

Overall, I think the strength of the paper is the simulated vertical structure and temporal variability of CH4 concentrations, which is representative of locations with greater water depth. It identifies the interesting role of thermal stratification in surface water, which can affect methane emissions, and the dynamics related to inflows from the North Sea. The authors discuss uncertainties in the overall budget, which could further be improved by considering methane production in the water column and by elaborating on the role of cold seeps and other point sources. Beyond simulating some interesting dynamics, I am currently not convinced that the model will be able to constrain a methane budget for the entire Baltic Sea in the future, as it will always be difficult to represent methane dynamics in coastal areas, and model output cannot be directly compared to measured profiles from particular locations. Maybe the authors can elaborate on whether switching to a 3-D model will be necessary.

*Response: It is indeed likely that the calibrated CH4 flux is mainly representative for present-day conditions. This is now discussed in Section 4, line 565-575 – see major point above.*

Specific comments:

Line 15: "land loads"

I do not understand what land loads could mean, since CH4 is not a solid but a gas. Based on the text, I think river runoff is meant. However, sometimes river runoff and land loads are mentioned in a single sentence as separate sources (e.g., line 594). In the text river runoff, river load, and land loads may denote the same source.

*Response: The "river runoff" indicates the freshwater supply to the various sub-basins, whereas "land load" includes both river loads and point sources of organic and inorganic carbon and nutrients. For methane, the contributions from point sources and river loads, respectively, are not known. For that reason, we have only included guessed river loads of methane in the model simulations. This has now been clarified in the text; "land load" of methane has been replaced by "river load" of methane.*

Lines 20-21: "to our knowledge this is the first time that CH4 isotopes have been included in a physical-biogeochemical model"

What about Nihous and Masutani (2006)? The full reference is listed below.

*Response: Thanks. Their model only includes fractionation during oxidation in the water column (and not air-sea exchange, etc.), but this is still a relevant reference that has now been included. The sentence has been modified and now reads (line 19-20): "Modeling of stable $CH_4$ isotopes can help to constrain process rates."*

Line 53-55: "This notion... Humborg et al. 2019)"

Could there not be an alternative explanation, such as increased methane production in the water column?

*Response: Potentially yes, but observations of gas flares indicate that outgassing from the sediments was the major source in this particular case. The sentence has nevertheless been rephrased and now reads (line 51-53): "This notion was qualitatively supported by acoustic observations of outgassing from the sediments during a recent field study, where exceptionally high CH4 emissions were reported from the coastal Baltic Sea at the end of a summer heat wave (~250 μmol m-2 day-1, Humborg et al., 2019)."*

Line 354: "calibrated"

This word is used several times in the text. However, due to the scarcity of data, I think it cannot be called a calibration. Also, it is annoying that the data is not shown. This shows the disadvantage of not using a 3-D resolved model.

*Response: The scarcity of data is a major issue for sure. For most of the parameters in the methane modelling we do not have observed rates to rely on, which means that the different rate constants are calibrated to produce simulated methane concentrations and isotope signatures as closely as possible to the few available observations. Perhaps this word can be used a little bit differently depending on context, but I still believe it is the most appropriate word in this case. I also doubt that using a 3D resolved model would make calibration easier considering the massive computational cost.*

Lines 430-431: "Furthermore, a local... model run."

This could indicate a local source of methane in the water column.

*Response: Yes. The potential influence from local $CH_4$ sources were already briefly addressed in Section 4 and Section 4.2. However, as suggested in one of the major comments, this has now been elaborated in some detail in Section 4, line 599-609 (see major comment above).*

Lines 482-486: "The rate constant... lakes cited above."

Rate constants could be compared to kinetic studies of methane oxidation in ocean water instead of lakes. It should be noted that microbes that oxidize methane may have more than

one trick on their sleeve (Rogener et al., 2018), allowing them to survive on other energy sources.

*Response: Text added in Section 4, line 583-589 (see major comment above).*

Line 599: "References"

The reference list is incomplete. At least, Weber et al. (2019) and Roth et al. (2022) are missing.

*Response: Thanks. These references (and several new references) have been added to the list.*

Minor comments:

Line 52-53: "In shallow... Borges et al. (2016)"

The sentence would improve by replacing "emissions" with "seafloor ebullition" and removing the last part.

*Response: Ok, updated according to suggestion (line 50-51):*

*"In shallow, organic-rich sediments, seafloor ebullition will increase in response to ocean warming due to increased biogenic $CH_4$ production and decreased $CH_4$ solubility (Borges et al., 2016)."*

Line 354: "intension"

*Response: Corrected.*

Line 388-389: "The δ13C-CH4 in water... temperature stratification."

The transition from the previous sentences is not smooth. It would be good to point the readers to the right figure here.

*Response: The text has been rewritten and now reads (483-487): "This temperature dependence on oxidation rates also has an impact on the isotopic composition of $CH_4$ – the $\delta^{13}C$-$CH_4$ in water above the top of the halocline is strongly influenced by the seasonality of temperature stratification (Fig. S3-S4, supporting information). However, the variations of isotopic composition in surface waters are significantly smaller than the variations at depth where $\delta^{13}C$-$CH_4$ mainly depends on transitions between oxic and anoxic conditions (Fig. 4-5)."*

Figure 6: Abbreviations should be explained, as figures should be understandable without reading the text. "ASE" is also not defined in the text.

*Response: Agreed. The figure legend has been updated; abbreviations have been replaced by words, and "land load" has been replaced by "river load".*

The term "redox zone" appears to be wrongly used at various locations in the text, where the authors intent to mean "redoxcline".

*Response: In this case we used the same wording as e.g., Schmale et al. (2012) and Jakobs et al. (2013;2014) for the oxic-anoxic transition zone. But anyway, "redox zone" has now been replaced by "redoxcline" in the text.*

While I would agree that the authors have by and large achieved their objectives, I think the manuscript in its current form won't meet the curiosity of interested readers, particularly for

those who are technical detail oriented and who are likely a significant fraction of gmd's readership. In the least, I think the authors should provide a thorough technique note that details the model structure with all governing equations and their supporting assumptions, as well as instructions on how initial and boundary conditions are set, and how the numerical solution is obtained.

For example, the current paper leaves me with many questions like:

- How is the reactive-transport problem being formulated?
- Does the model explicitly represent diagenesis?
- Is the sediment represented with explicit biogeochemistry?
- How diagenesis and biogeochemistry are coupled with temperature dynamics and vertical mixing?

*Response to the four points above: This model does not include a reactive-transport model for sediment diagenesis. Instead, we use a simplified parameterization where the model accounts for the depth dependent sediment pools of organic C, N, P, and Si at a resolution of 1 meter water depth. The total sediment areas at different water depths are described by the hypsographic functions for the respective basins. Sinking organic material from the water column feeds into these sediment pools at different depths, where the accumulated organic material is subjected to burial as well as temperature dependent mineralization. Mineralized carbon and nutrients can then again be released to the water column. Oxygen concentration in the water overlying the sediments controls the release of ammonium and nitrate, respectively, as well as the denitrification loss term. Oxygen concentration further controls phosphate sequestration, representing phosphate bound to iron oxides, as well as an additional release of phosphate to the water column during transitions from oxic to anoxic conditions, representing reduction of iron oxides.*

*The sediment dynamics in the model is described in some detail on line 212-228 in the revised manuscript.*

- In what way is river load applied?

*Response: River loads (as well as point sources and atmospheric depositions) of organic and inorganic carbon and nutrients are applied as monthly mean loads to the respective sub-basins (see line 232-233 in the revised manuscript).*

- And how the lateral and vertical transition of water depth and sediment thickness are handled?
- How is the lateral exchange formulated from the shallow water zone to deep water zone?

*Response to the two points above: The depth dependent areas of both water and sediment are described by the hypsographic functions for the respective sub-basins. Sediment thickness is*

*not explicitly modelled, instead, the model accounts for the depth dependent pools of organic C, N, P, and Si (see further in the response above, as well as the description on 212-228 in the revised manuscript). The model includes a parameterization for lateral transports of organic material from sediments at shallow depths toward deeper areas, representing resuspension and redeposition processes.*

- From some part of the paper, it seems the sediment is not explicitly represented. Then How should this be justified if the model is used for long-term projection, where active accumulation/degradation of sediment organic matter will be significant?

*Response: As discussed in responses above and also described in some detail in the updated model description (line 212-228), the sediment is indeed explicitly represented in the model. Active accumulation/degradation of sediment organic matter occurs in the model as a function on one hand on deposition rate, and on the other hand on burial as well as temperature dependent mineralization of organic material accumulated in the sediments. The model is a very useful tool for long-term projections and has in several prior publications been used both to do hindcast simulations and different future scenario simulations depending on climate change and nutrient load change.*

Besides, although the authors mentioned calibration in the paper, the results do not show any comparison with observations. (They did say the model more or less agree with some measurement in the text, but I think this is insufficient.) Since there is no differential equation of the reactive-transport system described, I cannot judge how well the model is performing, even I may trust the authors are making confident statement.

In all, I expect the authors do a major revision to present a more convincing manuscript to the readers.

*Response: Observed methane profiles were unfortunately not available. The "model description" section has been completely rewritten and now includes a lengthy qualitative description of hydrodynamic and biogeochemical processes in the model. The description also includes detailed references to studies where all governing equations and model parameterizations are formulated, but we have not repeated all the differential equations in the present paper – instead the focus here is on the new features of the model, i.e., the process parameterizations for stable isotope methane cycling, which is outlined in detail in Section 2.3.1-2.3.6.*

*The new and expanded model description now reads as follows, line 119-241 in the revised manuscript:*

[revised manuscript text omitted]

---

## Author Response (AR3)

Author's point-by-point response

The revised manuscript according to suggestions by the two referees.

**Response to referee #1:**

Comments by referee #1

Review of the revised manuscript: "Methane dynamics in the Baltic Sea: investigating concentration, flux, and isotopic composition patterns using the coupled physical, biogeochemical model BALTSEM-CH4 v1.0" by E. Gustafsson, B. G. Gustafsson, M. Hermans, C. Humborg, and C. Stranne, under review for Geoscientific Model Development

The authors have responded well to most review comments, and the manuscript improved substantially. The discussion about methane production in oxic water has been improved. The methane oxidation rates are compared to previous field studies, and in general, there is more consideration for previous biogeochemical research. The introduction became more informative. In response to the comments of the other reviewer, the model description has been expanded.

The most questionable part remains the overall methane budget for the Baltic Sea. The authors explain well that the model is not well-suited to constrain ebullitive emissions. The text also states: "Estimates by Weber et al. (2019) indicate that ebullitive fluxes contribute a major fraction of CH4 released to the atmosphere from shallow coastal areas" (lines 671-672). To be more precise, Weber et al. (2019) estimate that ebullitive fluxes account for roughly 50% of total global ocean methane emissions. Therefore, I agree with the following assessment: "This indicates that the simulated CH4 outgassing is likely underestimating the real outgassing from the Baltic Sea" (lines 679-680).

The average depth in the Baltic Sea is ~50 m. The model is fitted to data from the Gotland Basin, which has a maximum depth of ~460 m. Currently, it seems extremely difficult, if not impossible, to obtain empirical data needed to constrain a methane budget for the shallow parts of the Baltic Sea. Additionally, the model cannot resolve point sources and simulate methane ebullition. Coastal sediments are highly diverse in terms of organic matter loading, sedimentation rates, and substrates. These factors, which can only be distinguished for different locations by a 3-D modeling approach, significantly influence methane production rates. It is very likely that the methane production in shallow parts greatly differs from that in deeper parts.

My concern is that the model has been designed for deeper regions and fitted only to data from the Gotland Basin. Extrapolating these results to the entire Baltic Sea leads to highly unreliable estimates. Although the authors acknowledge the large uncertainties and describe their work as a preliminary budget, the current estimate is so uncertain that it does not enhance the quality of the paper. In the worst case, other studies may uncritically adopt these emission rates, potentially propagating significant errors.

In my opinion, the paper would be much stronger if it focused on the Gotland Basin, for which the model has been fitted. The simulated dynamics related to the interplay between physical and biogeochemical processes are sufficiently interesting. The budget for the entire Baltic Sea, on the other hand, is speculative at best. The text itself already contains many warnings. The authors may find it regrettable in the future if this part turns out to be significantly inaccurate.

I could agree with a minor revision if the quality of the paper is improved by reducing the emphasis on the preliminary budget. This could be achieved by removing the estimates of CH4 release from

sediments and CH4 emissions to the atmosphere for the entire Baltic Sea from the abstract and graphical abstract and instead providing values specifically for the Gotland Basin. Additionally, if the text focused more on the Gotland Basin, the analysis would be more robust, requiring less caution from the reader. The model output for the entire Baltic Sea could still be discussed in the text as part of ongoing model development, but should not be presented as robust scientific findings.

*Response: The Baltic Sea scale budget has now been replaced by a budget just for the Baltic Proper (including in addition to the Gotland Sea also the Bornholm and Arkona basins). Measured profiles of both CH4 concentration and isotopic composition are available not only for the Gotland basin, but also from the Bornholm and Arkona basin, indicating that the model (reasonably well) represents the central Baltic Sea area. On the other hand, the Gulfs of Bothnia, Finland, and Riga as well as the Kattegat basin are no longer included in the budget analyses. Furthermore, the graphical abstract was removed to reduce the emphasis on the preliminary budget calculations.*

*Consequently, the text has been modified on several places in the manuscript:*

- *The abstract has been updated and now reads:*

*"Methane (CH4) cycling in the Baltic Sea is studied through model simulations that incorporate the stable isotopes of CH4 (12C-CH4 and 13C-CH4) in a physical-biogeochemical model. A major uncertainty is that spatial and temporal variations of the sediment source are not well known. Further, the coarse spatial resolution prevents the model to resolve shallow-water near-shore areas for which measurements indicate occurrences of considerably higher CH4 concentrations and emissions compared to the open Baltic Sea. A preliminary CH4 budget for the central Baltic Sea (the Baltic Proper) identifies benthic release as the dominant CH4 source, which is largely balanced by oxidation in the water column and to a smaller degree by outgassing. The contributions from river loads and lateral exchange with adjacent areas are of marginal importance. Simulated total CH4 emissions from the Baltic Proper correspond to an average ~1.5 mmol CH4 m-2 y-1, which can be compared to a fitted sediment source of ~18 mmol CH4 m-2 y-1. A large-scale approach is used in this study, but the parametrizations and parameters presented here could also be implemented in models of near-shore areas where CH4 concentrations and fluxes are typically substantially larger and more variable. Currently, it is not known how important local shallow-water CH4 hotspots are compared to the open water outgassing in the Baltic Sea."*

- *The text on line 99-100 now reads:*

*"... 2. set up a preliminary CH4 budget for the Baltic Proper (where measured profiles of CH4 concentration and isotopic composition are available),..."*

- *The text on line 536-543 reads:*

*"To allow a preliminary assessment of the relative importance of different processes, total CH4 sources (river load, import from adjacent sub-basins, and sediment release) and sinks (outgassing, export to adjacent sub-basins, and pelagic oxidation) were aggregated over the Baltic Proper (sub-basin 7-9, Figure S1), representing the area where the model has been fitted based on available observations. The CH4 sources were largely dominated by benthic release which amounted to an average 4155 Mmol y-1 over the 2001-2020 period (Table 2). This source was mainly balanced by oxidation in the water column (3816 Mmol y-1, 92% of the sinks) and to a smaller degree by emission to the atmosphere (348 Mmol y-1, 8% of the sinks). The river load (11 Mmol y-1) and net exchange (import - export) with adjacent sub-basins (8 Mmol y-1) were comparatively small."*

- *Table 2 (line 546) has been updated with numbers for the Baltic Proper.*
- *The text on line 547-552 reads:*

*"Figure 6 illustrates simulated monthly fluxes, net accumulation as well as the total amount of CH4 in the Baltic Proper. The total CH4 stock amounted to almost 1800 Mmol over the ~2010-2014 period, which exceeded the stock before and after that period by a factor 3 (Fig. 6). This comparatively large CH4 stock was the result of a large anoxic deep-water volume and thus low oxidation rates (Fig. 2). There was an average net accumulation of 10 Mmol y-1 over the 2001-2020 period (Table 2), but net changes of the total CH4 stock between individual years varied considerably, which largely reflected oxygen dependent changes in CH4 oxidation rates (Fig. 6)."*

- *The original Figure 6 (line 554) has been replaced by a corresponding figure for the Baltic Proper.*
- *The text on line 559 reads*

*"This study presents a first quantification of key CH4 fluxes in the Baltic Proper"*

- *The text on line 698-701 reads:*

*"The calculated average total CH4 emission of 348 Mmol y-1 from the Baltic Proper corresponds to approximately 1.5 mmol CH4 m-2 y-1, and constitutes only about 8% of the fitted sediment source (~18 mmol CH4 m-2 y-1). The model includes both shallow- and deep water sediment areas, but the fitted sediment source is in the lower range of rates reported for a shallow-water coastal area (~21-34 mmol CH4 m-2 y-1; Roth et al., 2023),..."*

Minor comments

Lines 449-450: "the intention... Gotland Sea"

Please, make explicit what is meant by "existing observations" by spelling out the fitted parameters.

*Response: The sentence has been rewritten and now reads (line 449-451): "These parameters (i.e., CH4 oxidation rates and fractionation values, CH4 sources from the sediments, rivers, and the North Sea, as well as the isotopic compositions of these sources) are mostly fitted values, where the intention was to reasonably well reproduce existing observations of both CH4 concentration and isotopic composition from the Gotland Sea"*

Line 53: "250 µmol m-2 day-1":

Here µmol is used. In other parts of the manuscripts, both grams and moles are used. Please, use the same units throughout the manuscript.

*Response: Mole units are now consistently used for CH4 fluxes throughout the manuscript.*

"Calibration" versus "fitting" in the entire manuscript:

Fitting is more appropriate in the context of this study. There is a difference in meaning between the words. Fitting means adjusting the model to match known data as well as possible. Calibrating is a

broader process that includes fitting but also ensures the model performs well in different situations and against additional validation data.

*Response: The word "calibrated" has now been replaced by "fitted" throughout the manuscript.*

**Response to referee #2:**

Comments by referee #2

In the first paragraph, when describing the estimates of global CH4 emissions by the two approaches, can they authors include uncertainty range as well? The current description reads like too accurate.

*Response: The ranges have now been included on line 29-31: "... global CH4 emissions have been estimated to be 576 Tg CH4 y-1 (range 550-594), whereas bottom-up approaches (process-based modeling of land surface emissions and data on anthropogenic emissions) yield a total of 737 Tg CH4 y-1 (range 594-881; Saunois et al., 2020)"*